# The Global Long-term Microwave Vegetation Optical Depth Climate Archive VODCA

Leander Moesinger[1], Wouter Dorigo[1], Richard de Jeu[2], Robin van der Schalie[2], Tracy Scanlon[1], Irene Teubner[1], and Matthias Forkel[1]

[1]Technische Universität Wien, Department of Geodesy and Geoinformation, Gußhausstraße 27-29, 1040 Vienna, Austria
[2]VanderSat, Wilhelminastraat 43A, 2011 VK Haarlem, The Netherlands

**Correspondence:** Leander Moesinger (Leander.Moesinger@geo.tuwien.ac.at, vodca@geo.tuwien.ac.at)

**Abstract.** Since the late 1970s, spaceborne microwave radiometers have been providing measurements of radiation emitted by the Earth's surface. From these measurements it is possible to derive vegetation optical depth (VOD), a model-based indicator related to the density, biomass, and water content of vegetation. Because of its high temporal resolution and long availability, VOD can be used to monitor short- to long-term changes in vegetation. However, studying long-term VOD dynamics is gener-
ally hampered by the relatively short time span covered by the individual microwave sensors. This can potentially be overcome by merging multiple VOD products into a single climate data record. However, combining multiple sensors into a single product is challenging as systematic differences between input products like biases, different temporal and spatial resolutions and coverage need to be overcome.

Here, we present a new series of long-term VOD products, the VOD Climate Archive (VODCA). VODCA combines VOD re-
trievals that have been derived from multiple sensors (SSM/I, TMI, AMSR-E, Windsat and AMSR-2) using the Land Parameter Retrieval Model. We produce separate VOD products for microwave observations in different spectral bands, namely Ku-band (period 1987-2017), X-band (1997-2018) and C-band (2002-2018). In this way, our multi-band VOD products preserve the unique characteristics of each frequency with respect to the structural elements of the canopy. Our merging approach builds on an existing approach that is used to merge satellite products of surface soil moisture: First, the data sets are co-calibrated via
cumulative distribution function matching using AMSR-E as scaling reference. To do so, we apply a new matching technique that scales outliers more robustly than ordinary piece-wise linear interpolation. Second, we aggregate the data sets by taking the arithmetic mean between temporally overlapping observations of the scaled data.

The characteristics of VODCA are assessed for self-consistency and against other products. Using an autocorrelation analysis, we show that the merging of the multiple data sets successfully reduces the random error compared to the input data sets.
Spatio-temporal patterns and anomalies of the merged products show consistency between frequencies and with Leaf Area Index observations from the MODIS instrument as well as with Vegetation Continuous Fields from the AVHRR instruments. Long-term trends in Ku-Band VODCA shows that since 1987 there has been a decline in VOD in the tropics and in large parts of east-central and north Asia, while a substantial increase is observed in India, large parts of Australia, south Africa, southeastern China and central north America. In summary, VODCA shows vast potential for monitoring spatial-temporal
ecosystem changes as it is sensitive to vegetation water content and unaffected by cloud cover or high sun zenith angles. As

such it complements existing long-term optical indices of greenness and leaf area.

The VODCA products (Moesinger et al., 2019) are open access and available under Attribution 4.0 International at https://doi.org/10.5281/zenodo.2575599

*Copyright statement.* COPYRIGHSTATEMENTTEXT

# 1 Introduction

Vegetation attenuates microwave radiation that is emitted or reflected by the Earth surface. The degree of attenuation can be derived from passive and active microwave satellite observations and is commonly referred to as Vegetation Optical Depth (VOD) (Jackson and Schmugge, 1991; Vreugdenhil et al., 2016). The amount of attenuation depends on various factors, e.g.
the density, type and water content of vegetation, and the wavelength of the sensor (Jackson and Schmugge, 1991; Owe et al., 2008). Short wavelengths experience a higher attenuation by vegetation (and hence relate to higher VOD values) than longer ones (Liu et al., 2009; Owe et al., 2008; Kerr et al., 2018). As a consequence, VOD estimates from long wavelengths are generally more sensitive to deeper vegetation layers (e.g. stem biomass) while VOD estimates from short wavelengths are more sensitive to leaf moisture content (Chaparro et al., 2018; Tian et al., 2018; Fan et al., 2018; Konings et al., 2019). VOD increases
with the Vegetation Water Content (VWC) (Jackson and Schmugge, 1991) and therefore is related to the Above-Ground dry Biomass (AGB) (Liu et al., 2015) and its Relative Water Content (RWC) (Momen et al., 2017).

Satellite-derived VOD has a wide range of potential applications, including biomass monitoring (Liu et al., 2015; Brandt et al., 2018b), drought monitoring (Liu et al., 2018), phenology analyses (Jones et al., 2011) and estimating the likelihood of
wildfire occurrence (Fan et al., 2018; Forkel et al., 2017, 2019). VOD also correlates with various optical remote sensing indicators of vegetation greenness like Normalized Difference Vegetation Index (NDVI), Enhanced Vegetation Index, Normalized Difference Water Index (Grant et al., 2016), and Leaf Area Index (LAI) (Vreugdenhil et al., 2017) and hence also relates to plant productivity (Teubner et al., 2018, 2019). VOD has some distinct advantages over optical vegetation indexes for vegetation monitoring, such as a slower saturation and the resulting higher sensitivity to high biomass (Liu et al., 2015) or the ability
to be retrieved despite of cloud cover (Liu et al., 2011a) which are both advantageous for monitoring tropical forest regions (van Marle et al., 2016).

VOD products have been derived from multiple space-borne microwave sensors that have been in orbit since the late 1970s (Owe et al., 2008). These sensors have varying lifetimes and characteristics, resulting, e.g., from differences in microwave fre-
quency used, measurement incidence angles, orbit characteristics, radiometric quality, and spatial footprints. This complicates their joint use in studying long-term VOD dynamics. To overcome this issue, Liu et al. (2011a) proposed a long-term (1987-

2008) harmonized multi-sensor VOD data set by merging VOD products derived from the Special Sensor Microwave/Imager (SSM/I), the Microwave Imager onboard the Tropical Rainfall Measuring Mission (TMI), and the Advanced Microwave Scanning Radiometer - Earth Observing System (AMSR-E) through the Land Parameter Retrieval Method (LPRM; Owe et al. (2008)). Their methodology was inherited from the methodology used to produce the first long-term satellite-based climate data record of soil moisture within the the Climate Change Initiative of the European Space Agency (ESA CCI Soil moisture; (Dorigo et al., 2017, 2012; Liu et al., 2011c, 2012; Gruber et al., 2019)). In their methodology, all available observations were harmonized with respect to C-Band (6.9 GHz) VOD observations from AMSR-E, which was assumed to provide the highest quality observations (Liu et al., 2012). Only in periods where AMSR-E C-band observations were not available, other products were used instead. This approach ignores the fact that in a statistical sense a high-quality product can be fused with a low-quality product to create a product with a higher quality than either of the original products. This was systematically demonstrated for the merging of two level 2 soil moisture products (Gruber et al., 2017). Since the release of the multi-satellite VOD product by Liu et al. (2011a), significant progress has been made towards a better understanding of the VOD signal. It was shown that also the individual bands carry valuable information for different applications (Teubner et al., 2018; Chaparro et al., 2018), which demonstrates the need for frequency-specific VOD data sets. In addition, new sensors were launched, allowing the observational VOD records to be extended to the running present.

In this paper, we present a new series of long-term, harmonized VOD climate data records, called the VOD Climate Archive (VODCA), which are derived from multiple single-sensor level 2 products. VODCA uses a similar core methodology as in Liu et al. (2011a) and in ESA CCI Soil Moisture (Gruber et al., 2019) but incorporates the latest insights on VOD and climate data record production gathered during the last few years, and introduces recent satellite missions. We combine VOD observations from SSM/I, TMI, AMSR-E, WindSat, AMSR2 into global, harmonised long-term VOD products at a 0.25° spatial sampling and covering the period 1987-2018. First, we describe the input VOD data sets, followed by an overview of the fusion methodology. We then describe the main characteristics of the merged data sets in terms of spatial and temporal coverage and patterns, and their random error characteristics. We check the spatio-temporal characteristics for plausibility by comparing them to those of related satellited-derived biogeophysical products and complement the data set assessment by a trend analysis. We conclude the paper with a discussion on current limitations and ways forward.

## 2 Input data

### 2.1 Vegetation Optical Depth data sets

#### 2.1.1 The land-parameter retrieval model (LPRM)

LPRM v6 (van der Schalie et al., 2017; Owe et al., 2008; Meesters et al., 2005) is based on a radiative transfer model first proposed by Mo et al. (1982) and simultaneously retrieves soil moisture and VOD from vertical and horizontal polarized microwave data. The model assumes that the earth emits microwave radiation depending on its surface temperature $T_s$ and

emissivity $e$ which is a function of its dielectric constant $k$, which in turn is dependent on the surface soil moisture. Part of this radiation is then absorbed or scattered by water in the vegetation depending on its transmissivity $\Gamma$ and single scattering albedo $w$ while the vegetation itself also emits radiation depending on its temperature $T_v$. The resulting brightness temperature $T_b$ measured at the sensor can then be modeled as

$$T_{bp} = T_s e_p \Gamma + (1 - \Gamma) T_v (1 - w) + (1 - e_p)(1 - w) T_v (1 - \Gamma) \Gamma \tag{1}$$

where the subscript $p$ denotes either a vertical or horizontal polarization. Further, VOD ($\tau$) is related to $\Gamma$ and the incidence angle $u$ by:

$$\Gamma = \exp\left(\frac{-\tau}{\cos(u)}\right) \tag{2}$$

Since observations from the sensors used in this study are available in both horizontal and vertical polarization, eq. 1 is used to open a system of linear equations. While the the absolute measured $T_{bH}$ is lower than $T_{bV}$, it is more sensitive to changes in soil moisture while $T_{bV}$ is more sensitive to vegetation and surface soil temperature. This relationship in combination with the application of a separate retrieval algorithm to determine the temperature from 37-GHz vertical polarization measurements (Holmes et al., 2009) allows to solve the system analytically as described in Meesters et al. (2005).

The actual temperatures are difficult to estimate during daytime due to surface heating, while during nighttime, soil and vegetation are nearly in thermal equilibrium. This implies that nighttime retrievals are expected to have a lower temperature-related error than daytime retrievals (Owe et al., 2008). Therefore, to minimize error sources, only nighttime retrievals are used in VODCA. While LPRM v6 is not publicly available, older versions are available trough GFSC: https://disc.gsfc.nasa.gov/datasets/LPRM_AMSR2_D_SOILM3_001/summary

## 2.1.2 Sensor specifications

The used VOD data sets were derived from brightness temperature measurements of various space-borne sensors active since 1987 (Fig. 1).

The Advanced Microwave Scanning Radiometer (AMSR-E) onboard Aqua retrieved microwave observations from 2002 to 2011 in six bands, of which we only consider the C-, X-, and Ku-band. Their spatial footprint is $75 \times 43$ km, $51 \times 29$ km and $27 \times 16$ km respectively. AQUA is on an sun-synchronous circular orbit, passing the equator at 1:30 PM ascending and 1:30 AM descending mode (Knowles et al., 2006; Kawanishi et al., 2003).

The Advanced Microwave Scanning Radiometer 2 (AMSR2) is an improved version of AMSR-E onboard GCOM-W1 continuing AMSR-E's measurements since 2012 with similar bands, orbit and overpass times but with a slightly higher spatial resolution: $62 \times 35$ km, $42 \times 24$ km and $22 \times 14$ km, for C-, X-, and Ku-band respectively. In addition, AMSR2 also contains a second C-band (7.3 GHz) that can be used to cover areas where Radio-Frequency Interference (RFI) is present in the primary

C-band channel (6.9 GHz) (Meier et al., 2018). During preliminary analysis, we discovered that the AMSR2 Ku-band VOD retrievals have an apparent break in late 2017. Since then, the values observed are globally systematically lower than before, indicating possibly a calibration error in Ku-band brightness temperatures. While the exact reasons are unknown to us, until the matter is resolved we do not include Ku-band data after 2017-08-01 into VODCA. This shortens the Ku-band VOD product by 16 months. VOD retrievals from X- and C-band AMSR2 seem unaffected and are used until the end of 2018.

The Special Sensor Microwave Imager (SSM/I) is onboard a series of DMSP satellites. We use the VOD data retrieved from F-8, F-11 and F-13. From the 7 available bands of SSM/I we use only VODfrom Ku-band which has a resolution of $69 \times 43$ km. The equatorial crossing time varies between the DMSP satellites, but all are on sun-synchronous orbits (Wentz, 1997).

Among other sensors, the Tropical Rainfall Measuring Mission (TRMM) carried the TRMM Microwave Imager (TMI). TRMM is the only satellite used which has a non-near-polar orbit with an inclination of 35 degrees. Up to 2001 it had an altitude of 350 km, which then got boosted to 400 km leading to a slight decrease in spatial resolution. TMI was active from 1997 to 2015. Of the 9 channels we only use its X- and Ku-band, which have a spatial resolution of $63 \times 37$ km / $72 \times 43$ km and $30 \times 18$ km / $35 \times 21$ km pre/post boost, respectively (Kummerow et al., 1998).

WindSat onboard Coriolis was launched in 2003 on a sun-synchronous orbit providing radiometric measurements in five bands, of which the C-, X- and Ku-band were used to derive VOD. The spatial resolution is $39 \times 71$ km, $25 \times 38$ km and $16 \times 27$ km. Due to some periods of non-operation, WindSat contains temporal data gaps (Gaiser et al., 2004). Unfortunately we were unable to gain access to data past July 2012, even though WindSat is still operational.

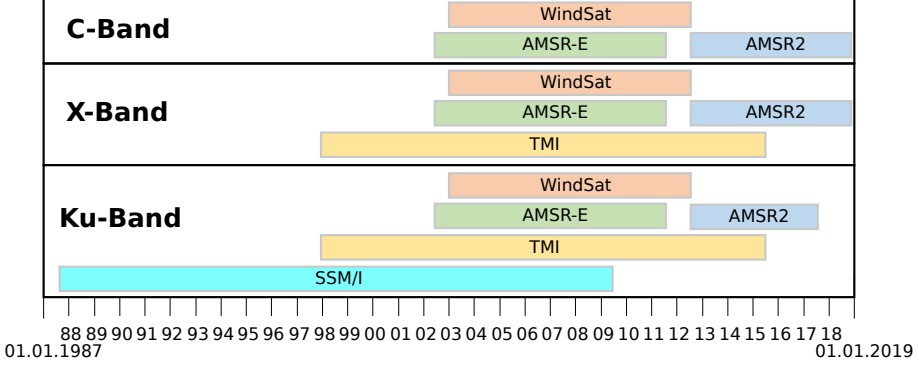

**Figure 1.** Time periods of the sensors used for each band.

**Table 1.** The input VOD data sets used with their temporal coverage, local ascending equatorial crossing times (AETC) and used frequencies [GHz] for each product.

| Sensor | Time period used | AECT | C-Band | X-Band | Ku-Band | Reference |
|--------|-----------------|------|--------|--------|---------|-----------|
| AMSR-E | Jun 2002 - Oct 2011 | 13:30 | 6.93 | 10.65 | 18.7 | van der Schalie et al. (2017) |
| AMSR2 | Jul 2012 - Jan 2019 | 13:30 | 6.93 & 7.3 | 10.65 | 18.7 | van der Schalie et al. (2017) |
| SSM/I F08 | Jul 1987 - Dec 1991 | 18:15 | | | 19.35 | Owe et al. (2008) |
| SSM/I F11 | Dec 1991 - May 1995 | 17:00 - 18:15 | | | 19.35 | Owe et al. (2008) |
| SSM/I F13 | May 1995 - Apr 2009 | 17:45 - 18:40 | | | 19.35 | Owe et al. (2008) |
| TMI | Dec 1997 - Apr 2015 | Asynchronous | | 10.65 | 19.35 | Owe et al. (2008) |
| WindSat | Feb 2003 - Jul 2012 | 18:00 | 6.8 | 10.7 | 18.7 | Owe et al. (2008) |

## 2.2 Evaluation data

### 2.2.1 Vegetation Optical Depth product from Liu et al, VOD_Liu

We compared the VODCA datasets with the previously created multi-sensor, multi-band VOD dataset (Liu et al., 2011b, 2015), hereafter called VOD_Liu. VOD_Liu covers the period January 1993 to December 2012 and is based on VOD retrieved via

LPRM from SSM/I (Ku-band), TMI (X-band), and AMSR-E (C/X-Band) observations. The values are scaled to AMSR-E and methods are in place to fill gaps due to frozen ground and to correct for large-scale open water bodies. We expect that the data that are publicly available were subject to some temporal smoothing since the data are mostly gap-free. The smoothing is not described in Liu et al. (2011b, 2015) and supplementary information.

### 2.2.2 MODIS leaf area index

To verify the plausibility of VODCA we compare it to MODIS leaf area index (LAI), MOD15A2H version 6 (Myneni et al., 2015). LAI is the ratio of one-sided leaf area to ground area and is estimated from the solar-reflective MODIS bands using a look-up-table based approach with a back-up algorithm that uses empirical relationships between NDVI, LAI and fraction of photosynthetically active radiation (FPAR). Field studies with different crop types showed that VOD is closely related to LAI (Sawada et al., 2016), a relationship that has been already used to assess VOD products derived from active sensors

(Vreugdenhil et al., 2017). The data is available globally since 2002 with an 8-day temporal resolution, and is for comparison purposes spatially downsampled from its native resolution of 500 metres to a quarter degree grid. The original data are available on https://lpdaac.usgs.gov/data_access/data_pool .

### 2.2.3 AVHRR vegetation continuous fields

We use the vegetation continuous fields (VCF) version 1 derived from data of Advanced Very High Resolution Radiometer

(AVHRR) instruments (Hansen and Song, 2018; Song et al., 2018). The VCF product shows the fractional cover of bare ground,

short vegetation and tree canopy, where trees are defined as all vegetation taller than 5 metres in height and short vegetation is defined as vegetation smaller than 5 metres. VCFs are provided as yearly files from 1982 to 2016 indicating the fractional coverage during the local annual peak of growing season. The VCF product is distributed by the Land Long Term Data Record (LTDR) at https://search.earthdata.nasa.gov

Given the relation of VOD with vegetation height and biomass Giardina et al. (2018); Liu et al. (2015), it seems sensible to assume that VOD would increase from areas with bare ground and short vegetation to areas with high tree cover. Hence, we use the VCF data to calculate the mean VCF from 2002 to 2016 and compare it to the mean of the VODCA products from 2002-2017 (sec. 4.1). Furthermore, we calculate the VCF trends from 1987 to 2016 and compare it to the trends in the merged

Ku-band VOD over the same period (sec. 4.4.2). Song et al. (2018) also calculated VCF trends by first determining whether there is a significant trend with a Mann-Kendall test and then calculating the slope with a Theil-Sen estimator. Both are non-parametric tests that are robust to outliers, but using different methods to mask for significance and estimate the slope can lead to significant slopes that are still very small. To avoid this issue, we also calculate the slope using a Theil-Sen estimator and use the Theil-Sen estimator to determine a 95% confidence interval for the slope and remove any slopes where the zero-slope

is within the confidence interval.

## 3   Methods

For each of the VODCA products, we use almost exactly the same methodology. Exceptions to this common methodology are described at the end of the respective subsection. Each product is computed without any influence of the others. The main difference between the three products is the time period spanned, resulting from the varying availability of input data (Fig.

1). The fusion process involves three main processing steps: First, preprocessing involves masking for spurious observations and spatial and temporal collocation of the data sets. Second, bias between the different sensors is removed by scaling them to AMSR-E VOD. Ultimately, the collocated and bias-corrected observations of all data sets are merged in time and space.

### 3.1   Preprocessing

Level 2 VOD data in swath geometry were first projected onto a common regular $0.25° \times 0.25°$ latitude-longitude grid using

nearest neighbour resampling. The different sensors visit the same spot on the Earth surface at different times of the day. To facilitate further processing, we do not take into account the exact time of observation. Instead, we selected for every UTC midnight the closest nighttime value in a window of $\pm 12$ hours which is identically as in the ESA CCI soil moisture processor (Dorigo et al., 2017). Since in subsequent processing steps the values of different sensors with different measurement times will be merged, one can consider the resulting values as nightly averages.

Basic masking operations were applied to remove potentially spurious observations. Specifically we mask for RFI, low land surface temperatures (LST), and VOD values $\leq 0$ as follows:

– RFI: Artificial microwave emitters on the Earth's surface distort the signal received by the satellite, causing the resulting VOD values at those locations to be unreliable. RFI is typically frequency-specific. RFI flags were already provided with the level 2 VOD data and were based on de Nijs et al. (2015). Any observations affected by RFI are removed.

– LST: Due to the different dielectric properties of ice and water, reliable retrievals can only be made if the ground is not frozen. Therefore, we remove observations where the LST is below 0° C. Masking for LST was based on the temperature retrievals of from Ka-band (Holmes et al., 2009), which is found on all the multi-channel instruments used in VODCA, and were provided with the level 2 VOD data.

– Negative VOD values: VOD retrievals <0 are physically impossible and are therefore removed from the data sets. We also remove VOD values of 0.0 (floating point zero). The reasoning is twofold: First, it is physically only possible to get floating point zero VOD if there is virtually no vegetation, making it very unlikely for most parts of the globe. Second, we observed that this case occurs surprisingly often also in non-desert regions, and that these values never fitted well with the other observations. This indicates that most VOD values of zero are artifacts that have to be removed.

The above masking is applied independently to all sensors and bands. A special case is AMSR2, which has two channels in C-band, i.e. at 6.9 and 7.3 GHz. If possible, the observations from the 6.9 GHz band are used, but if the observation in this channel is masked, the 7.3 GHz observation is used instead (if unmasked) to fill gaps. This only causes a minor reduction in quality, as the two C-bands are strongly correlated (Supplementary Figure 1). A flag indicating the channel ultimately used in the merged data set for each observation is provided in the metadata.

## 3.2 Cumulative distribution function (CDF) matching

We use a new implementation of the CDF-matching technique based on a combination of piece-wise linear interpolation and linear least squares regression. CDF-matching is used to correct for systematic differences between the VOD values of each sensor, which may result e.g. from the individual sensor designs, incidence angles, spatial footprints and the slight differences in the frequencies used. The goal of CDF-matching is to scale a source data set such that its empirical CDF becomes similar to the empirical CDF of the reference data set. CDF-matching is applied on a per-pixel basis and has been successfully used for similar tasks that require the correction of higher order differences between data sets (Liu et al., 2009, 2011a, 2012; Dorigo et al., 2017).

### 3.2.1 Improvements to ordinary piece-wise linear CDF-matching

Ordinary piece-wise linear CDF-matching (Liu et al., 2009, 2011a; Dorigo et al., 2017) predicts for each [0, 5, 10, 20, 30, ... , 80, 90, 95, 100] percentile of the source data the same percentile of the reference data set. Values between the $n^{th}$ and $n^{th}$+1 percentile are then scaled using linear interpolation. While the scaling parameters are determined only from temporally

overlapping observations, during prediction there can be values outside the training range. These values are scaled by extrapolating the first or last percentile interval. This method preserves the ranks of the source and computes rather fast. However, the first and last percentiles are defined by the lowest and highest observations, respectively, in both source and reference time series. Hence, a single outlier can greatly affect the parameters of these percentiles, making them unreliable. We propose here
improvements to this method to derive more robust scaling parameters that are not specific to VOD data but rather should be generally applicable in similar situations.

The first improvement is to fit a linear model using the sorted observations smaller than the second percentile with an intercept through the second percentile. This gives more reliable scaling parameters for low values since all the data between
the lowest and second-lowest percentiles are used instead of just the lowest value. In case a different number of observations exists in the source and reference, the data with less observations is padded by linear interpolation during training. In a similar fashion, a model is fitted for observations above the penultimate percentile.

We further increase the robustness of the CDF-matching parameters by dynamically increasing the step size of the percentiles if only few observations are available. The number of observations varies greatly from grid point to grid point and from sensor
to sensor. If too few observations exist between two subsequent matching-percentiles (a "bin"), the CDF-matching may overfit, leading to unreliable parameters. To counteract this, we dynamically reduce the number of bins and increase the size of the bins based on the number of observations.

### 3.2.2 Stability of parameters

To evaluate whether the new matching technique is more robust to outliers than the original piecewise linear CDF matching
method, we simulate the variances of the derived parameters of each bin for a varying number of training observations using artificial values. The use of artificial values allows us to test the method without being influenced by the artifacts inherent to real data. To achieve this, we sample a set of source and reference values from a standard normal distribution, and then determine the resulting CDF-matching parameters. For each evenly spaced percentile bin, we determine the slope in radians. This is repeated a few thousand times for various numbers of values (representing time series with a varying number of observations),
each time drawing new values. If a CDF-matching method is robust, the determined slopes should have low variance due to the values always being drawn from the same distribution.

We run this both with piece-wise linear CDF-matching and our new method. However, for this simulation we do not dynamically decrease the number of bins, as we are solely interested in the performance of the linear regression scaling the first and
last percentile. Both methods are tested with the same randomly drawn data.

The resulting variances in the slope, for each percentile bin, for both methods, depend on the number of observations used for the parameter determination. This is shown in Fig. 2. The results in the middle bins are exactly the same, as the same methodology in used for these bins. However, in the case of linear piece-wise interpolation, the slope parameters of the first

and last bins have a much higher variance than the middle bins as they are affected by outliers. In contrast, the slopes determined by the least-squares method have a much lower variance. In both cases we can also see that the more observations we have, the lower the variance of the slope parameter is, showcasing the reasoning behind reducing the number of bins dynamically if too few observations are available.

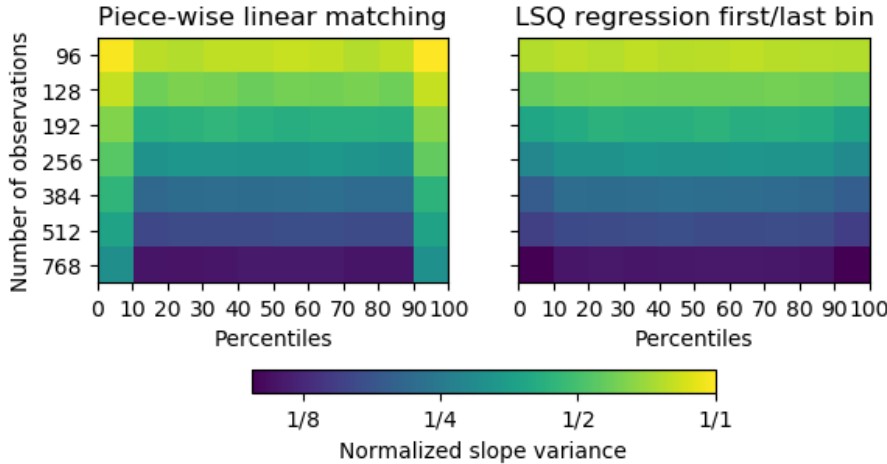

**Figure 2.** Variance of the derived slope, depending on the number of observations and the percentile bin for both piece-wise linear CDF-matching techniques. The color is log normalized.

### 3.2.3 Practical implementation

While there is no "true" reference to scale to, AMSR-E has almost global coverage and has a long temporal overlap with all other sensors but AMSR2. Hence, the empirical CDFs of WindSat, TMI and SSM/I are directly scaled to the one of AMSR-E, similar to Dorigo et al. (2015). To preserve any potential trends in both source and reference data, only dates when both have a valid observation are used. If at a certain location less than 20 temporal overlapping observations exist, no reliable scaling parameters can be determined and the source time series is dropped. A bin size of 20 was chosen as compromise between data coverage and often used bin sizes. A bin size of 50 observations is often used as a rule of thumb for univariate regression to get robust estimates (Green, 1991). However, our main goal was rather to prevent time series with very few observations from learning spurious scaling parameters and we also did not want to loose all time series with less than 50 values. As such 20 was chosen as a compromise.

AMSR2 does not share any temporal overlap with AMSR-E and therefore cannot be directly scaled based on overlapping observations. Instead, for X- and Ku-band, scaled observations of TMI can potentially be used to bridge this gap. This is done according to the following logic: If possible, AMSR2 is scaled to the rescaled TMI. Should there not be enough overlapping observations, the scaling parameters are determined from all observations of the first two years of AMSR2 and the last two

years of AMSR-E. While this removes any potential trends in the first two years of the AMSR2 period, these trends are still assumed to be smaller than the removed bias. Last, if there are also not enough AMSR2 or AMSR-E observations available in those years, the whole AMSR2 time series is dropped. For C-band, which is not covered by TMI, the AMSR2 data are always matched directly to AMSR-E by using the last and first two years of both sensors. The published data sets contain a flag indicating the matching method, allowing the user to remove the AMSR2 observations matched directly to AMSR-E if desired.

Since the scaling parameters are determined using only a subset of all observations, during prediction there can be values outside the training range. The regression is not forced to go through the origin, therefore if the predicted values can potentially be smaller than 0. These values are deemed unreliable and are removed. However, this occurs very rarely, only a fraction of about $1/10^6$ to $1/10^8$ of values are lost this way.

## 3.3 Merging

For all bands, the CDF-matched time series of all individual sensors are merged into a single long continuous time series. For a certain pixel at a certain time step, three possible scenarios can occur:

1. If on a certain date no sensor has an observation, a data gap will result in the final product;

2. If only one sensor has an observation, the CDF-matched value will be directly integrated in the final product;

3. If multiple sensors have an observation on a certain date, their arithmetic mean is taken.

This means that the number of sensors contributing to each observation within a time series can vary greatly. For each observation in the final product there is a flag indicating which sensors have contributed to it. Although more sophisticated weighted merging methods based on least squares have been proposed to merge multiple satellite observations (Gruber et al., 2017, 2019), estimating these weights, i.e. indicators of the relative quality of the individual data sets, is a non-trivial task. This particularly applies to VOD, for which no appropriate independent reference data exist. However, in most cases, the arithmetic mean appears to be a robust approximation of optimal merging (Liu et al., 2012).

Alternatively, one could also take the median instead of the mean. This would likely be more robust to outliers but would only make a difference if three or more concurrent values exist. As such the difference would likely be very small and thus this is not explored in detail.

## 4 Properties of the long-term Vegetation Optical Depth data sets

### 4.1 Spatial patterns and temporal dynamics

Figure 3 shows an example of X-band VOD time series in Austria at different stages of merging procedure together with MODIS LAI. The original VOD time series have visible systematic differences between each sensor. The CDF-matched VOD

time series have been scaled to AMSR-E and visually do not show systematic differences between sensors. The statistical distributions of VOD from the sensors are similar after matching (Fig. 3 b). This example grid point is north of 38N and thus outside the spatial coverage of TMI, therefore AMSR2 has been scaled to AMSR-E directly using non-temporally overlapping observations. The merged VOD time series shows comparable seasonal dynamics like LAI.

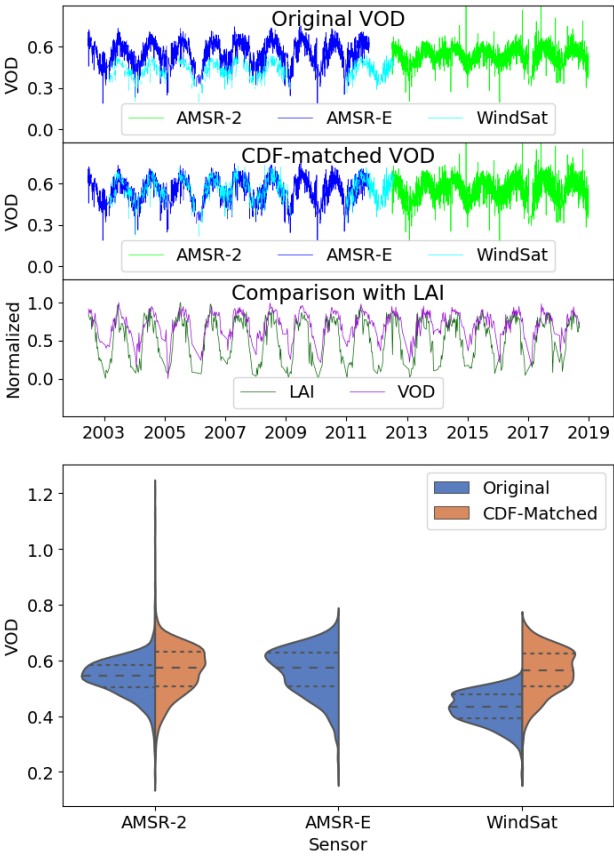

**Figure 3.** Top: Example X-band time series for a grid cell in Austria (15.125°E, 48.125°N, mostly farmland with about 20% forest) at different processing steps. Time series of the original VOD data of all available sensors are shown in the top panel, the CDF-matched series in the middle panel, and the final merged VOD (VODCA) is shown together with MODIS LAI in the bottom panel. In the bottom panel VOD and LAI are both normalized, VOD is downsampled by moving average to match the temporal 8-day resolution of LAI. Bottom: Violin plot showing the effect of CDF-matching on the statistical distribution of VOD.

The global spatial patterns of average VOD between June 2002 and June 2017 is shown for each band in Figure 4 (a-c). This period was selected because all bands have global coverage in this time period. All bands show similar spatial patterns, matching the ones of the VCF land covers (Fig. 4 (e), with high VOD in tropical and northern forests and lower VOD in grassland and desert regions. The same pattern is also visible in canopy height (Simard et al., 2011) and MODIS LAI (Fig. 4

(d), even though the LAI in the tropical forests is much higher than in the boreal forests, while VOD is similarly high in both regions. Based on the principle that the penetration of microwaves increases with wavelength, the maximum VOD is highest at shorter wavelengths (Ku-band) and smallest at longer wavelengths (C-band). This can also be seen in Figure 4 (f) which shows the average VOD of each band for locations dominated by high tree cover (vegetation height > 5m), short vegetation (< 5m) or bare ground. Similarly to previous findings based on L-band (Konings and Gentine, 2017), this figure also shows that on average VOD is highest in forests and lowest over bare ground.

The temporal dynamics of VOD across different latitudes shows plausible seasonal patterns of vegetation phenology (Fig. 5). In general, summer months have the highest VOD: in the Tropics and Subtropics due to increased precipitation during that time, while in northern/southern regions due to the increased temperature and consequent vegetation growth and (leaf) biomass gain.

The VOD time series do not show any visible artificial breaks, indicating that the biases have overall been successfully removed from the individual sensors before merging. To make potential artificial breaks more visible, we investigated the seasonal anomalies per latitude (Fig. 6). The anomalies are calculated by collecting all the observations of a latitude, calculating the monthly mean, subtracting the multi-year monthly average and removing any potential linear trends using ordinary least squares regression. Hence the anomalies should either represent natural variability or artifacts due to shifts in available sensors. In the latter case, one would expect global anomalies to be visible either due to bias or differing spatial extent.

Most anomalies are limited in both space and time, their start or end does not coincide with a change in sensors and indicating that they are due to natural causes. Anomalies in MODIS LAI show similar patterns like VODCA anomalies, showing that surface events manifest in both in a similar way. The VOD_Liu anomalies are very similar to the VODCA anomalies, the biggest difference being that the texture is less coarse due to the temporal smoothing present.

To further assess the stability of VODCA, the correlation of VOD with LAI was calculated for different blending periods similar as in Dorigo et al. (2015) (Figure 7). The blending periods are chosen for each band such that each period corresponds to a different set of input sensors (Figure 1) and that each period is long enough to calculate reliable coefficients. Both the correlation between the raw time series as well as the anomalies indicate that the temporal dynamics are consistent over the whole length of the time series.

## 4.2 Spatio-temporal coverage

The temporal and spatial coverage of the merged VOD time series for each band is shown in Figure 8. The coverage of the merged products is defined by the spatial and temporal coverage of sensors (Fig. 1). For any band in any time span with at least one sensor, most parts of the globe have for at least 40% of all days an observation, while in any time period with at least two sensors about 70% of all days have a valid observation. TMI is the only sensor with a non-polar orbit of 35°N/S, leading to an increased coverage in that region in Ku- and X-band from 1997-2015. The latitude affects the coverage in multiple ways: Northern regions are generally more often covered by the polar-orbiting satellites but on the other hand, frozen grounds and

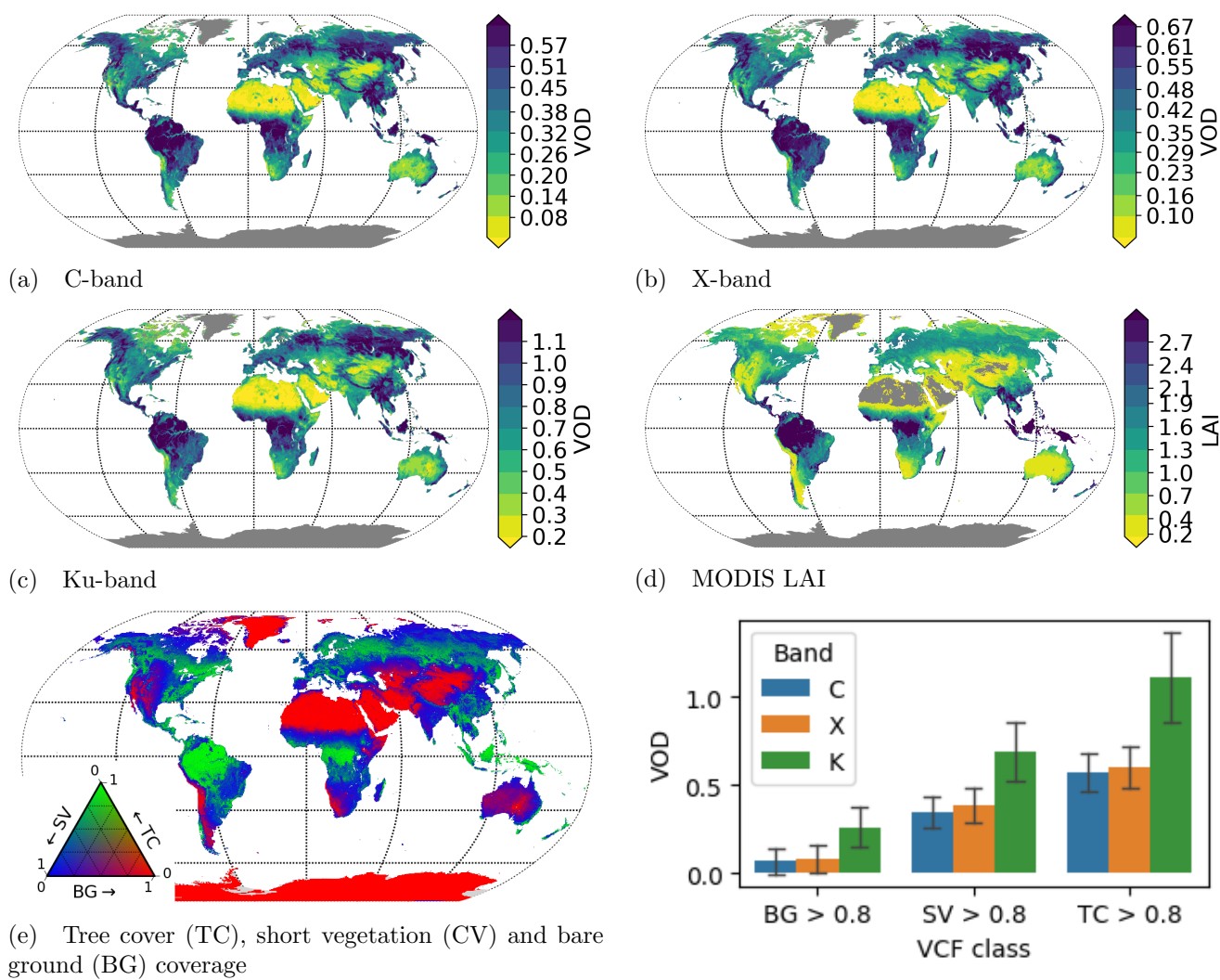

(a)   C-band

(b)   X-band

(c)   Ku-band

(d)   MODIS LAI

(e)   Tree cover (TC), short vegetation (CV) and bare ground (BG) coverage

(f)   Mean VOD of each band depending on VCF class

**Figure 4.** Global spatial patterns of average multi-sensor VOD from each band (2002 to 2017), average MODIS LAI (2002 to 2017) and average VCF (2002-2016 and distribution of VOD for locations with high tree cover (TC), short vegetation (SV) and bare ground (BG) greater than 0.8. The error bars indicate the standard deviation within each group.

snow cover inhibit the retrieval of VOD in winter. The low coverage band near 23°N is the result of LPRM not converging on a valid solution in very arid regions due to the extreme soil dielectric constants in these regions (de Jeu et al., 2014).

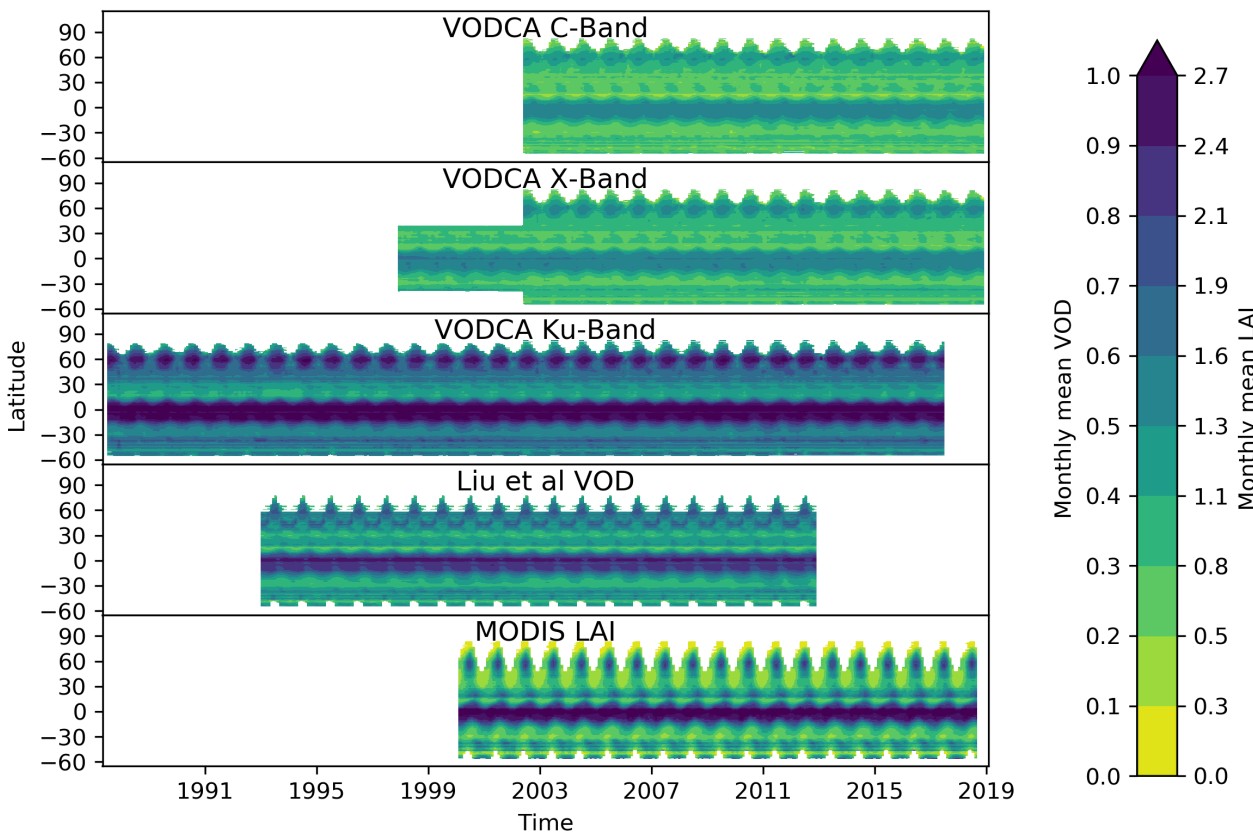

**Figure 5.** Hovmoeller diagrams showing the monthly mean VOD per latitude for each Band of VODCA, VOD_Liu and for LAI

In some locations the merged VOD products have fewer observations than in the original products. This data loss can be caused by a failure of the merging procedure, in detail explained in section 3.2.3. Matching failures are always a result of insufficient AMSR-E data and hence the data loss occurs in similar regions for all sensors of one band. The lack of AMSR-E data is in most cases due to either RFI or low temperatures in mountainous regions. As an example Figure 9 shows, for all
5  bands, where the CDF-matching failed for WindSat data. Ku-band is the least affected (Fig. 9 (c)), where only about 2% of the grid points are lost, mostly in the Himalayas. In X-band the matching fails for about 5% of the grid points, mostly in large parts of the Sahara (Fig. 9 (b)). C-band is most affected by data loss (10%), mostly in some parts of the USA where additional RFI prevents accurate retrievals (Fig. 9 (a)) (Njoku et al., 2005).

## 4.3   Random error characteristics

10  To validate the performance of our merging approach we evaluate the change in autocorrelation as an indicator for precision. Merging overlapping observations from multiple sensors is supposed to result in data that has a higher precision than the data of any of the individual sensors. But without a higher-quality external reference data set, assessing the change in precision is

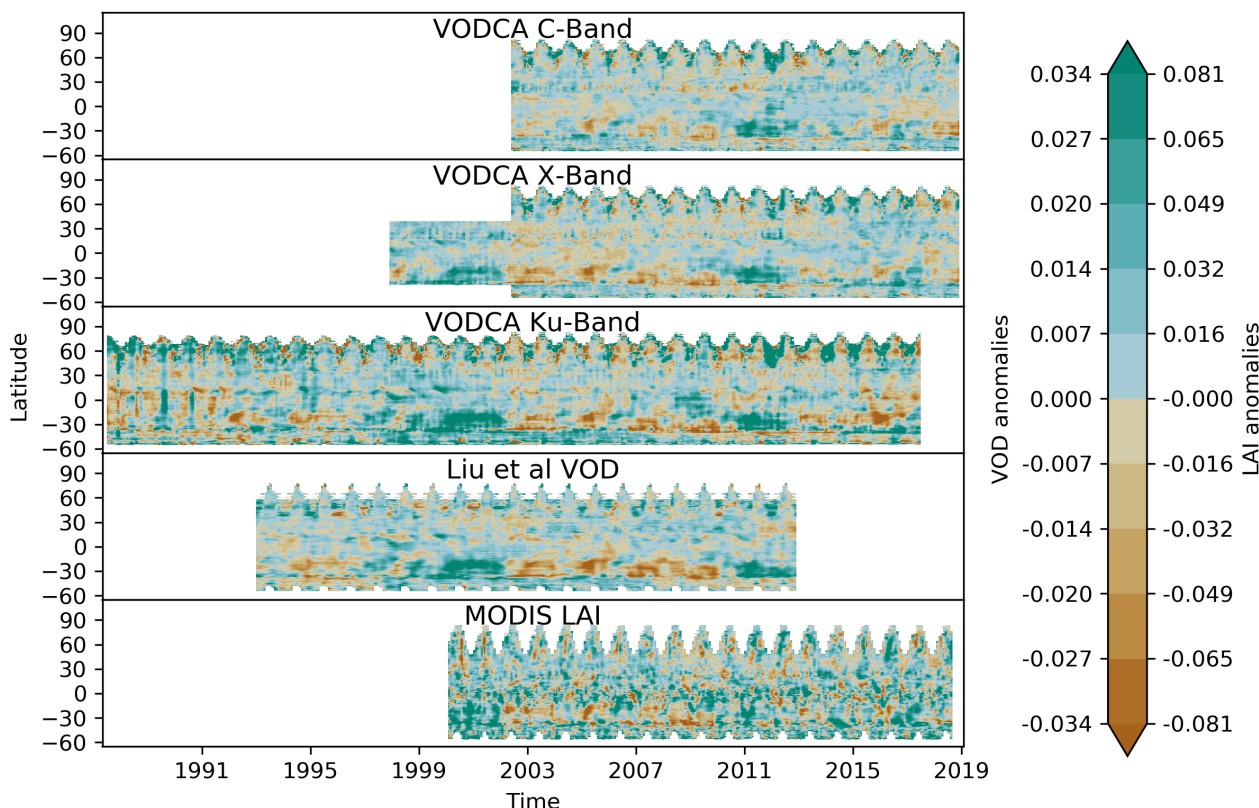

**Figure 6.** Hovmoeller diagrams showing anomalies of the monthly means per latitude for each band of VODCA, VOD_Liu and for LAI

non-trivial. However, we can assume that there is supposed to be a high degree of temporal autocorrelation between subsequent observations because VOD is related to gradual changes in plant water content and biomass (Momen et al., 2017; Konings et al., 2016). Therefore we calculated the difference between the first-order temporal autocorrelation before and after merging. The autocorrelation coefficient is strongly dependent on the temporal resolution. As seen in sec. 4.2, the temporal resolution of VODCA increases if multiple sensors are available. Therefore directly comparing the autocorrelation coefficients between the individual sensors and the merged products would lead to an increase in autocorrelation that is related to the temporal resolution rather than to the precision. Therefore the temporal resolution is kept unchanged by using only observation dates existing both in the pre-merge and post-merge data set.

The autocorrelation differences for X-band are shown in Figure 10. The other bands show similar results and are available in the supplementary Figures 1-3. The autocorrelation of the merged time series is on average higher than the autocorrelation of the input series, indicating an overall decrease in noise. However, sometimes the gain in autocorrelation of one sensor mirrors the loss of the autocorrelation of the other, likely due to the former sensor being more noisy than the latter, e.g. in Alaska or

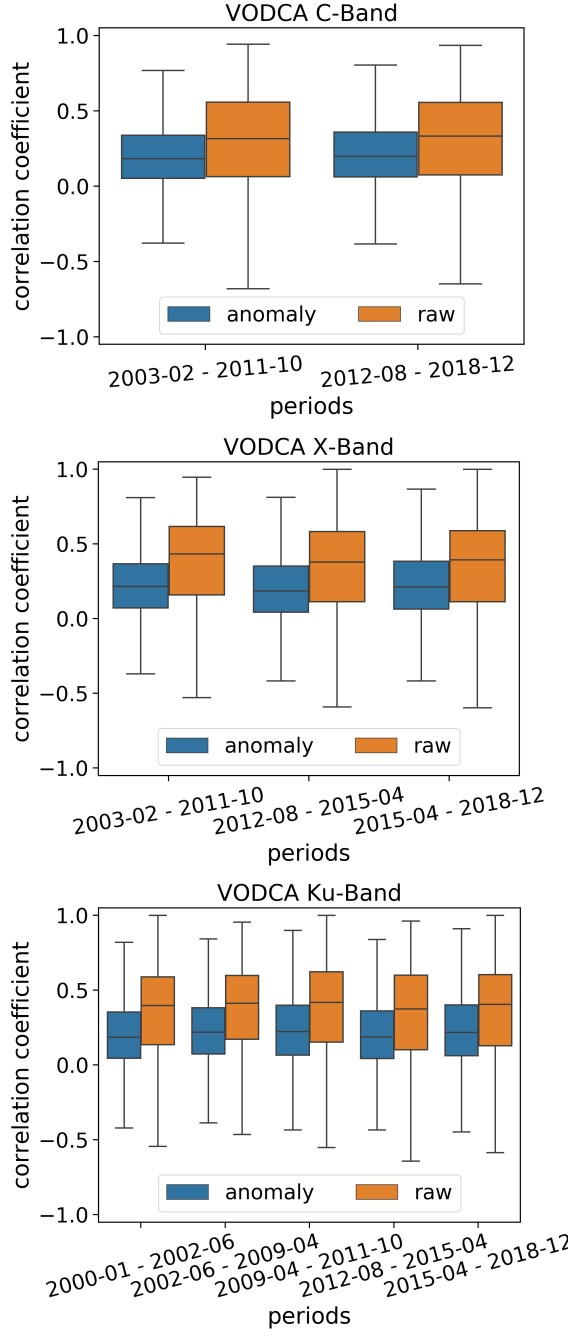

**Figure 7.** Correlation between VODCA and MODIS LAI, raw time series and anomalies, for different blending periods.

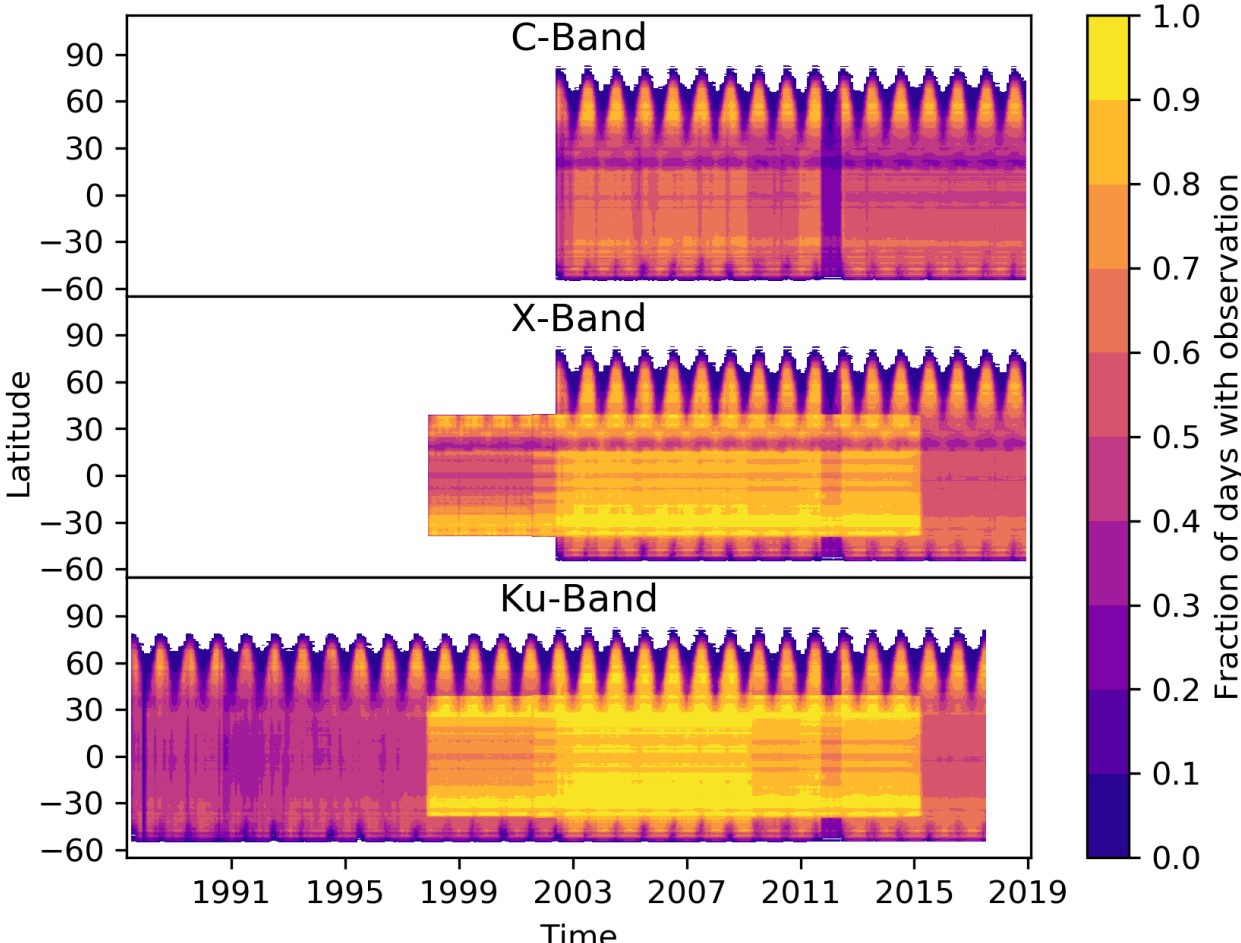

**Figure 8.** Hovmoeller diagrams showing for each latitude and month the fraction of days per month with observations. The number of observations of a latitude and month are counted and then divided by the number of days per month and the number of land grid points at that latitude.

east Russia in X-band of AMSR-E vs. WindSat. This means that locally, sometimes a single sensor has a higher precision than VODCA. But there are also regions where the merged VOD autocorrelation is higher than any of the input time series, e.g in Europe or central north America. This is likely to occur when all sensors have a similar precision, meaning that none of them is dragging the precision of the others down.

5    A noteworthy case is TMI where the autocorrelation of the merged time series is almost always higher. This could mean that the TMI data is very noisy and is dragging the overall quality of the merged data down. We investigated this possibility by experimentally not including TMI in VODCA. This resulted in average in a lower gain in autocorrelation for the other data

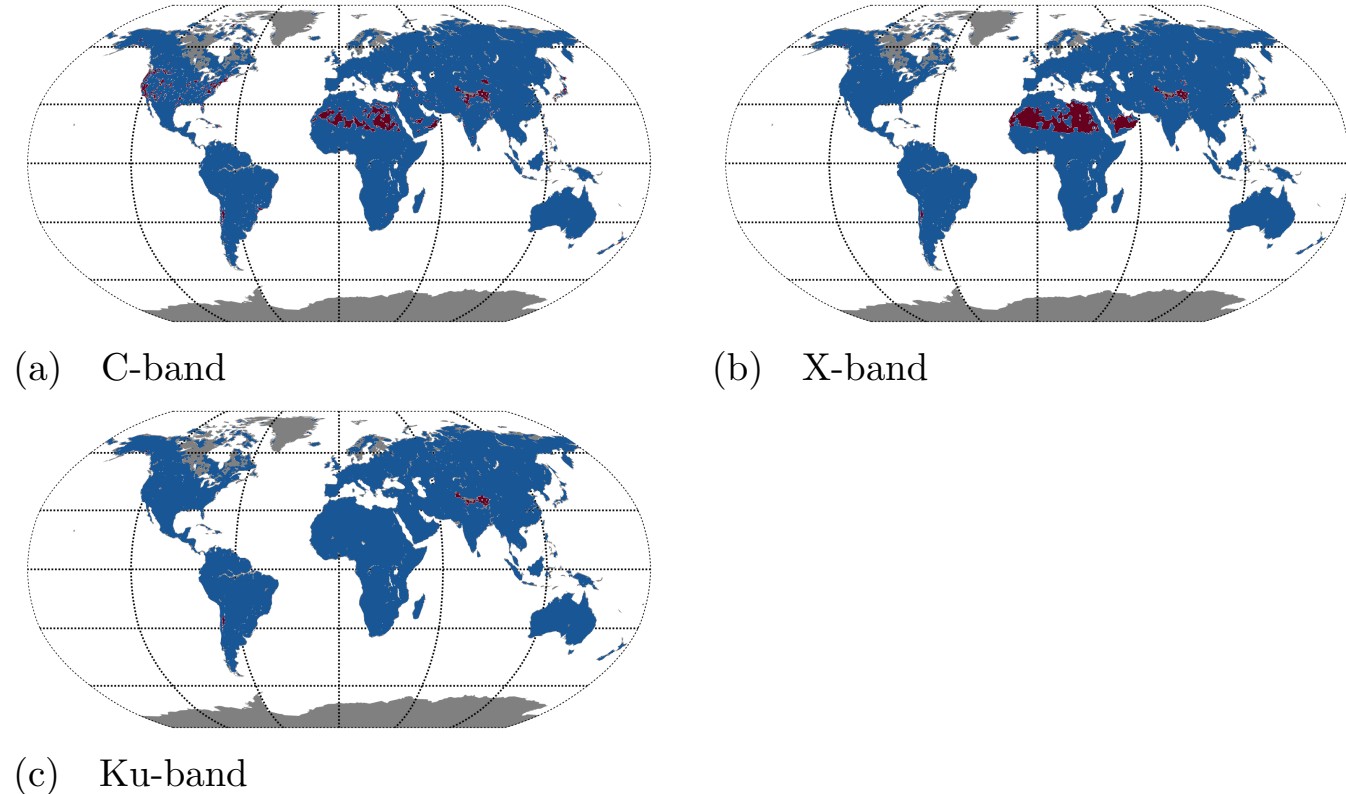

(a)   C-band

(b)   X-band

(c)   Ku-band

**Figure 9.** Data loss of during CDF-matching of different WindSat bands. CDF-matching failed for the red grid points and therefore the data of WindSat at that location is dropped. Very similar looking maps exist for the other sensors in the supplement Fig. 4

sets, indicating that the TMI data is still positively contributing to the precision of the merged products by reducing the noise of the end product.

## 4.4   Comparison of VODCA with LAI, VOD_Liu and Vegetation Continuous Fields

### 4.4.1   Correlation between Vegetation Optical Depth and LAI

5   A direct validation of VODCA is not possible because of the lack of appropriate in situ measurements. Hence it is only possible to assess dynamics in VOD with dynamics in related variables such as LAI or land cover. Globally, LAI and VODCA time series and their seasonal anomalies are positively correlated over large areas (Figure 11). For all bands, the highest correlations with LAI can be found in grassland-dominated regions such as in African Savannahs, Australia and in parts of South America. Correlations are usually lower in forested regions and even slightly negative in parts of tropical forests such as in

10   the Amazon. The negative correlations in tropical forests could be caused by drought periods where vegetation water content and hence VOD should decline but LAI possibly increases (Myneni et al., 2007; Saleska et al., 2007), although a green-up of

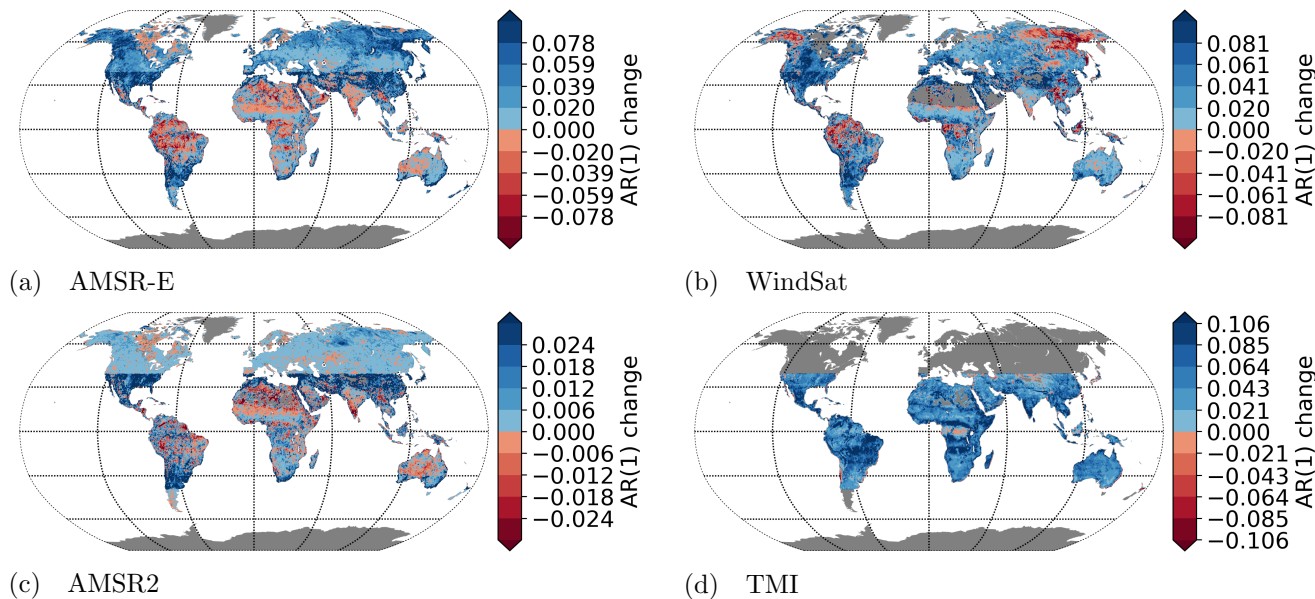

**Figure 10.** First-order auto-correlation change due to merging of X-band data for each sensor.

the Amazon under drought is highly debated (Samanta et al., 2010, 2012; Morton et al., 2014). However, this comparison of VODCA and LAI demonstrates that VODCA reflects plausible seasonal and short-term changes in vegetation and will likely provide additional information on vegetation dynamics on top of LAI and other related optical biophysical vegetation products from optical remote sensing.

To assess differences between the temporal dynamics of VODCA and VOD_Liu, we compared both to MODIS LAI. Because VOD_Liu is temporally smoothed, comparing daily values is inadequate. Instead, we first resample both datasets to monthly averages and calculate the Spearman correlation to the also monthly averaged MODIS LAI, only using dates existing in all datasets. The downsampling leads to slightly higher correlation coefficients (Figure 12) than using the daily values (Figure 11)

10 due to decreased noise, while the spatial patterns stay the same. The highest correlation has VODCA X-band, with a global average of 0.42, followed by VODCA Ku- and C-band with 0.39 and 0.37 respectively. Lowest in average is VOD_Liu with 0.33. It could be that the lower correlation is a result of being a mix of multiple bands or because the VODCA products use more input datasets, resulting in more accurate values. Either way, this indicates that the VODCA products capture temporal dynamics better.

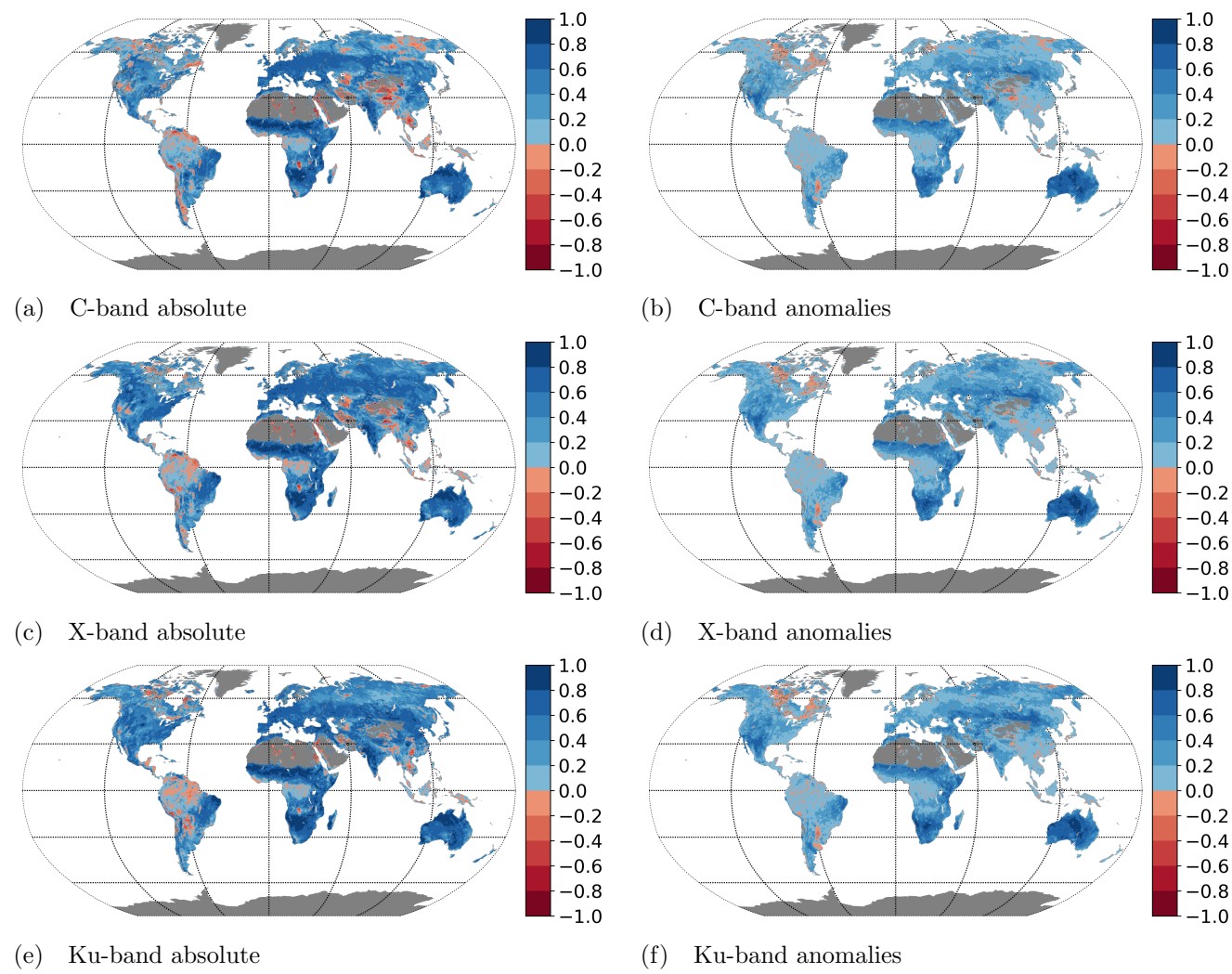

**Figure 11.** Spearman correlation coefficient between VODCA VOD and MODIS LAI for each band. The left column shows the correlation for the absolute signal, the right column for the anomalies from the long-term VOD climatology.

### 4.4.2 Trend-analysis of VODCA, VOD_Liu, LAI and Vegetation Continuous Fields

To evaluate the relationship between C-, X-, Ku-band VODCA, VOD_Liu, MODIS LAI and VCF changes and to gain a first insight into the long-term changes in VOD we assess linear trends in the data sets. Yearly averages are used to determine the trends and their confidence intervals via the Theil–Sen estimator. Trends whose upper and lower confidence interval do not have the same sign or either of them is zero are regarded as non-significant and are not displayed in Figures 13, 15 and 14 . Figures 13 (a-c) show the C-, X- and Ku-band VODCA trends from 2002-06-19 to 2017-06-19 during which all bands have

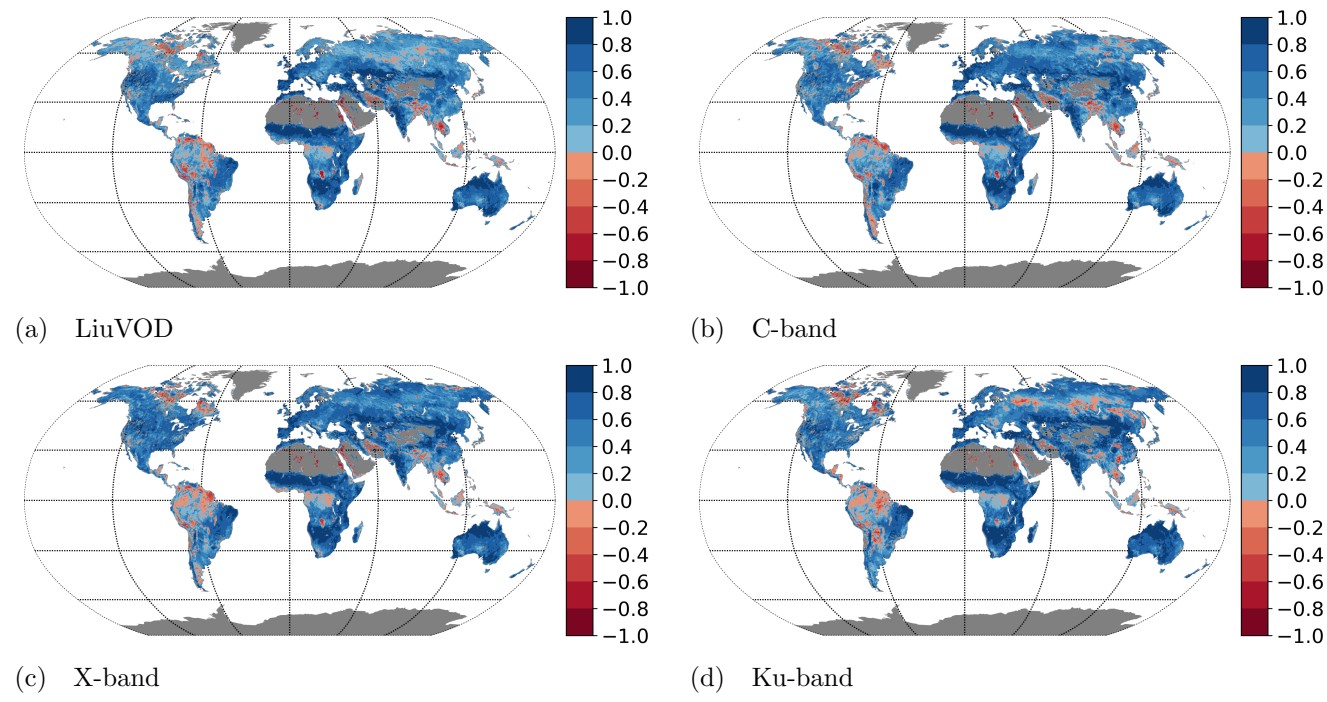

**Figure 12.** Correlation of monthly VOD_Liu and the VODCA products with MODIS LAI. For this analysis, the data are first resampled to monthly averages, then only the months where all four data sets have values are used.

global coverage. The trends are visually very similar in all bands, confirmed by the spatial Spearman correlation coefficients of 0.88 between the C- and X-band trends, 0.89 between C- and Ku-band and 0.91 between X- and Ku-band, calculated using only locations where both bands have a significant trend. This further reinforces that all bands react very similarly to vegetation changes. The spatial overlap of trends is shown in Figure 13 (d), where each location is classified based on the sum of posi-

5   tive and the sum of negative trends. Locations with no significant trend in any band are not displayed. The three classes with contradicting trends (1|1, 2|1, 1|2) are rare as together they make up only $4.2\%$ of the displayed points. Conversely, $48\%$ of the land points are covered by the four classes with at least two agreeing trend directions without any contradicting trend (2|0, 3|0, 0|2, 0|3). The agreement in trends between frequencies indicates that the longer Ku-band series can be used as indicator of the shorter X- and C-band series in trend analyses. Further, the LAI trends of the same time period (Fig. 13 (e)) match the VOD

10  trends very well overall, even though in detail the strength and location of the trends vary.

The trends of Ku-band VODCA and VOD_Liu were determined (Figure 14) to asses whether studies that have been analyzing VOD trends using VOD_Liu would get different results if they were repeated using VODCA Ku-band instead. Ku-band VODCA is used because it has the longest overlap with VOD_Liu (1993 to 2012).

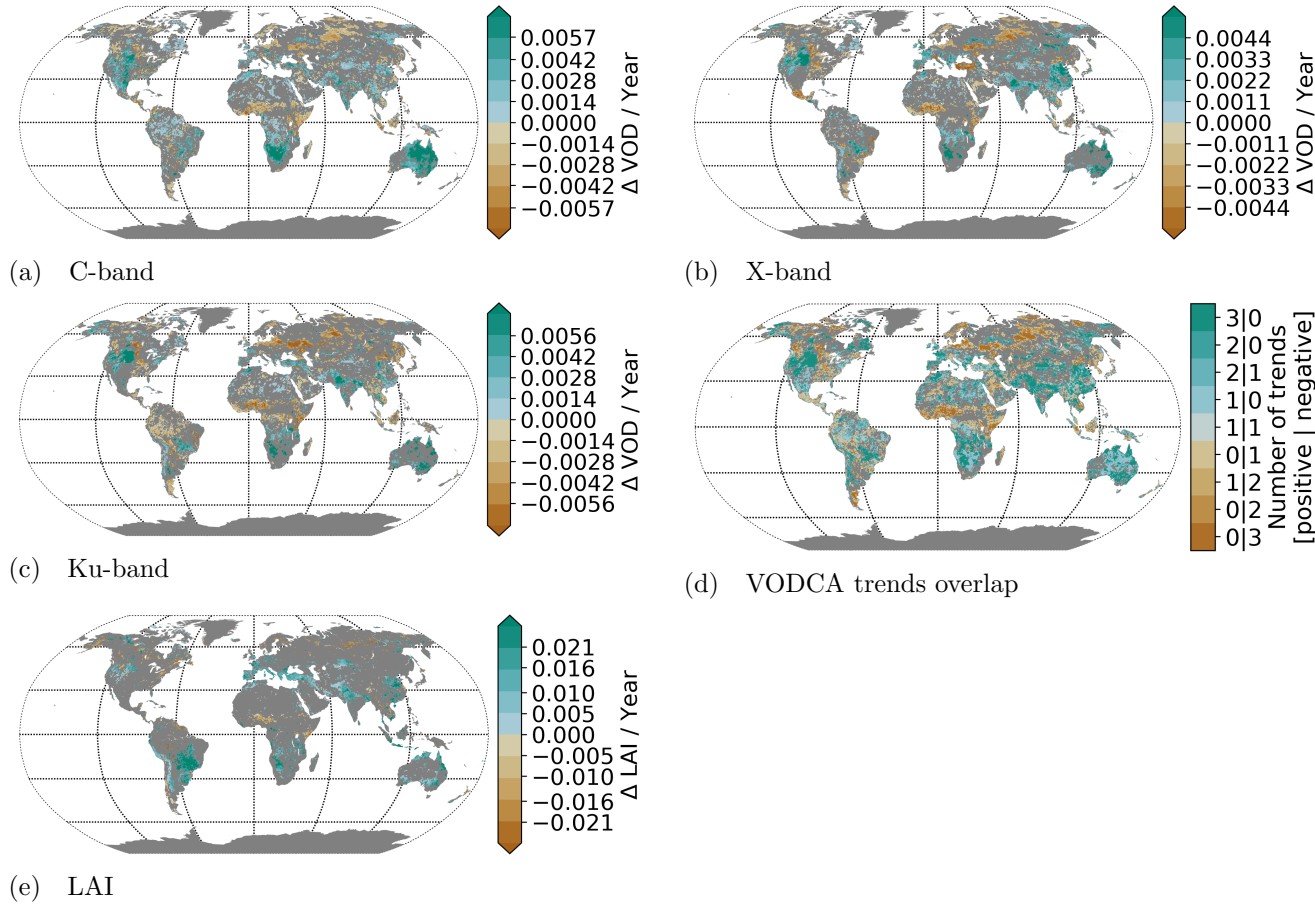

(a) C-band

(b) X-band

(c) Ku-band

(d) VODCA trends overlap

(e) LAI

**Figure 13.** Trends of various bands between 2002 to 2017 of VOD (a-c) and LAI (e). Non-significant trends are not displayed, the trends are calculated by Theil-Sen regression using yearly mean values. Figure (d) shows trend classes based on the number of VOD bands showing a positive|negative trend. Their order and color are indicative of the likelihood of the trend.

On a global scale, we see the almost exact same patterns in both VOD series, therefore studies performed at that scale would get similar results for both data sets. However, on a local scale the patterns differ simethimes; E.g. in most of Turkey Ku-band VODCA shows an increase, while VOD_Liu shows a decrease in VOD. As such regional studies might get case-by-case very different results depending on which dataset is used.

Taking advantage of the much longer length of the Ku-band, another trend analysis is done for this band using the data from 1987 - 2016 (Fig. 15 (g)) to give a first impression of the changes within the last thirty years. Overall we see a decline in VOD in the tropics, likely due to deforestation, and in large parts of Mongolia, attributed to variations in rainfall and surface

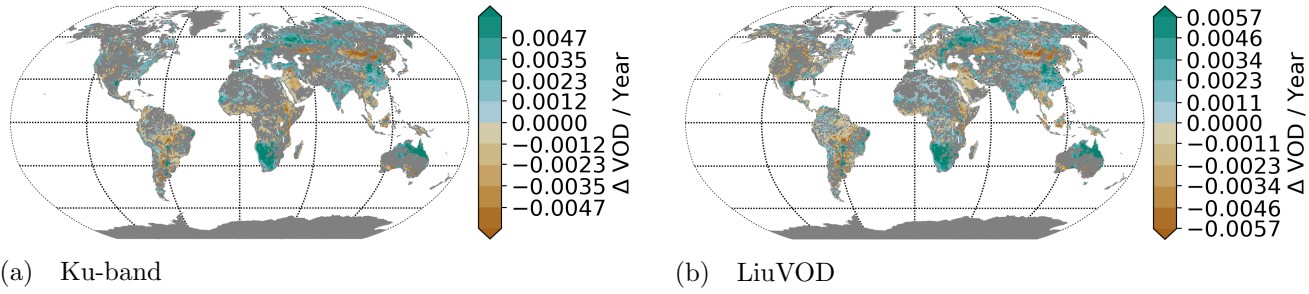

(a)   Ku-band

(b)   LiuVOD

**Figure 14.** Trends between 1993 to 2012 of Ku-band VODCA (a) and VOD_Liu (b). Non-significant trends are not displayed, the trends are calculated by Theil-Sen regression using yearly mean values.

temperatures as well as increased life stock farming and wild fires (Liu et al., 2013). VOD increased strongly in India and large parts of China, mostly due to an increase in croplands in the former case and due to both an increase in forest and croplands in the latter (Chen et al., 2019). VOD also increased in northern parts of Australia, matching trends in FPAR and precipitation seen in Donohue et al. (2009). Other regions with increasing VOD are south Africa and central north America. Of a question-
able nature is the wide spread positive trend in the Sahara given LPRMs struggle to retrieve VOD here. Most of the changes observed for VOD are mirrored in the VCF changes from 1987 to 2016 (Fig. 15 (f), see sec. 2.2.3 for details). The large bare ground losses in India, China and the north African shrubland manifest as positive VOD trends. Likewise, the deforestation in south America and land degradation with hotspots in Mongolia, Afghanistan or southwestern USA coincide with a loss in VOD. Also the patterns of tree cover gain in eastern Europe and European Russia coincide with increased VOD. While there
do not seem to be any areas where VOD and VCF contradict each other clearly, some trends are only visible in one of the data sets. For example the strong increase in VOD in southern Africa cannot be observed in VCF.

## 5   Current limitations and possible improvements

### 5.1   AMSR2 scaling to TMI

Upon closer inspection of the trends in Figure 13, we can see in north America a spatial break in X- and Ku-band trends at 35°N. North of this latitude AMSR2 data of 2012-2014 were matched to the AMSR-E data of 2010-2012, while south of this line temporally overlapping scaled TMI values were used to bridge the gap between the two sensors. Unusual low VOD values can be observed in this region in the years 2012 to 2015 in both X-and Ku-band. This indicates that the CDF-matching does not correct the bias between the sensors but artificially removes the difference that is due to surface processes. Consequently,
the matched AMSR2 data has a slight positive bias north of 35°N in large parts of north America. For users we advise to be careful when using X- and Ku-band values after July 2012 north/south of 35$^{o}$N/S as well as C-band values after July 2012

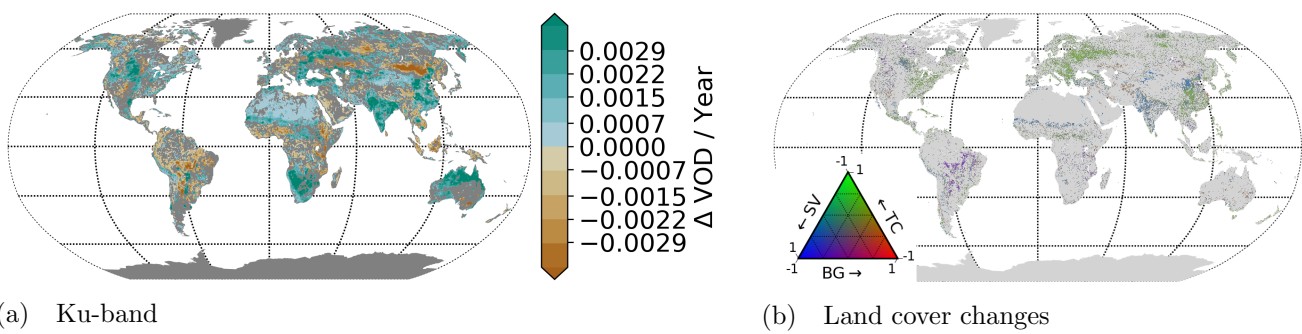

(a)  Ku-band                                              (b)  Land cover changes

**Figure 15.** Trends between 1987 to 2016 of Ku-band VOD (a) and VCF tree canopy, short vegetation and bare ground (b). Non-significant trends are not displayed, the trends are calculated by Theil-Sen estimator using yearly mean values.

globally as the AMSR2 data might induce a bias. Currently there exists a flag indicating how AMSR2 has been CDF-matched. With ongoing AMSR-E vs. AMSR2 Level 1 intercalibration efforts by JAXA we expect to reduce spurious observations in the AMSR2 period in th enear future.

### 5.2   Data loss while CDF-matching

As described earlier, CDF-matching failed because of missing AMSR-E data in some regions, mostly in the Himalayas (Fig. 9). One possible solution to avoid this data loss would be to substitute the CDF-matching parameters of these locations with the parameters from locations with similar dynamics in VOD. This could be done by clustering the time series and using the parameters of another location within the same cluster. Taking this one step further, one could also investigate the possibility of using all the data in one cluster to derive a single set of CDF-matching parameters and use these to scale all the source time

series within it. Not only would this allow to scale all the data without loss, but the increased number of values available for each parameter determination would also lead to more robust CDF-parameters. However, generating meaningful clusters from hundreds of thousands long time series containing missing values while keeping the computational cost at bay is anything but trivial (e.g. Mikalsen et al. (2018)). Besides, even though clusters may be composed of time series with very similar characteristics, the VOD signal at each location may still have its unique features resulting e.g., from land surface characteristics or

vegetation species composition.

### 5.3   Data gaps in the input data sets leading to increased noise

Averaging multiple temporally overlapping observations reduces noise (sec. 4.3). However, this can be only done if overlapping observations exist. While theoretically the maximum number of observations is defined by the number of available sensors, 20  in practise usually fewer observations are available due to gaps in the individual time series. Hence, filling short gaps in the

original time series of each sensor could potentially increase the precision of VODCA. Since VOD changes slowly over time (Konings et al., 2016), it is intuitively clear that even if a sensor has no valid observation on a certain date, the value is expected to be similar to the value of the dates before and after. Therefore one could fill short gaps with a model that at least implicitly uses autocorrelation for its predictions, such as gaussian processes (Camps-valls et al., 2017).

## 5.4  L-band product

An L-band product would be of great use to the scientific community, as L-band VOD has been instrumental in analyzing vegetation patterns (e.g. Brandt et al. (2018a); Tian et al. (2018); Brandt et al. (2018b); Chaparro et al. (2018)). Although we produced an experimental L-band product based on LPRM-SMAP and LPRM-SMOS using the same methodologies as for the other bands, the evaluation of this L-band product showed that it is not yet fit for release for a number of reasons. First, merging SMOS and SMAP does not result in a time series that is longer than just SMOS alone, therefore in terms of temporal extent nothing is gained. Second, the temporal coverage is highly unbalanced, with the SMAP period having a much higher density. This carries the high risk that users might apply unfitting methods to the data. Third, the autocorrelation analysis indicated that VODCA-L has a higher level of noise than pure LPRM-SMAP. Nevertheless, given the great scientific interest in L-band VOD, we continue working on a VODCA L-band product. Yet, a lot of work is still required, such as assessing the impact of the VOD retrieval algorithms (e.g., LPRM SMAP and SMOS (van der Schalie et al., 2017; Owe et al., 2008; Meesters et al., 2005), SMOS-IC (Fernandez-Moran et al., 2017) and MT-DCA (Konings et al., 2016)), and developing more suitable merging algorithms that can deal with the low temporal variability of L-band VOD compared to the other frequencies.

## 5.5  Effect of merging different observation times and geometries

Litarature has shown that the observation time has an influence on the retrieved VOD (Konings and Gentine, 2017; Konings et al., 2017) and that the spatial footprint and resampling method and the resmpling reference time affect the quality of merged soil moisture products (Dorigo et al., 2015). However, overall very little knowledge currently exists about the effect of mixing observation times and sensor geometries (incidence angles, spatial footprint,...) of multiple VOD values. Further research on these topics would improve the understanding of VOD and may lead to more advanced merging procedures that take these effects into account.

## 6  Conclusions

In this paper we presented VODCA, three long-term VOD data sets spanning up to three decades that can be used in studies of the biosphere. We were able to remove most of the biases between the different input sensors by co-calibrating them to AMSR-E. The merging leads to observations with less noise than the input data sets. The trends of the different VODCA products (C-, X-, Ku-band) correlate very strongly with each other and show similar spatial distributions and temporal dynamics as trends in LAI and VCF. Compared to the latter products, which are based on solar-reflective remote sensing, VOD has the benefit of being unaffected by cloud cover, allowing generally for more than 40% of days having a valid VOD observation. A major

ongoing issue is the potential bias in AMSR2 due to no temporally overlapping observations with other sensors. Future efforts will focus on resolving this and other issues while future VODCA releases will continuously update the climate archive with recent observations.

*Data availability.* The VODCA products (Moesinger et al., 2019) are open access (Attribution 4.0 International) and available at Zenodo
https://doi.org/10.5281/zenodo.2575599

*Author contributions.* Wouter Dorigo, Leander Moesinger, and Matthias Forkel designed the study. Leander Moesinger performed the analyses and wrote the manuscript together with Matthias Forkel and Wouter Dorigo. All authors contributed to discussions about the methods and results and provided feedback on the manuscript.

*Competing interests.* The authors declare that they have no conflict of interest

*Acknowledgements.* The MODIS LAI data (MOD15A2H, v006) by Myneni et al. (2015) were retrieved from the online data pool, courtesy of the NASA EOSDIS Land Processes Distributed Active Archive Center (LP DAAC), USGS/Earth Resources Observation and Science (EROS) Center, Sioux Falls, South Dakota, https://lpdaac.usgs.gov/data_access/data_pool .

The VCF annual data (VCF5KYR, v001) by Hansen and Song (2018) were retrieved from the online NASA Earthdata Search, courtesy
of the NASA EOSDIS Land Processes Distributed Active Archive Center (LP DAAC), USGS/Earth Resources Observation and Science (EROS) Center, Sioux Falls, South Dakota, https://search.earthdata.nasa.gov .

The authors acknowledge the TU Wien University Library for financial support through its Open Access Funding Program.

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
