# Peer review of "The Global Long-term Microwave Vegetation Optical Depth Climate Archive VODCA"

_Earth System Science Data, 2019_

## Short Comment (SC1) · 26 Apr 2019

As a byproduct of microwave-based surface soil moisture (SSM) retrieval, vegetation optical depth (VOD) is closely related to total vegetation water content (VWC) which is an effective estimator of above-ground biomass (AGB). Time series of VOD dataset serves as an important supplement to vegetation change studies which are previously based exclusively on optical sensor data such as GIMMS3g NDVI. One of the biggest concerns for such a time series dataset generation is the issue of temporal consistency. This article proposes an improved Cumulative Distribution Function (CDF) matching technique inherited from CCI SSM generation, and provides a 30-year (maximum and to be elongated with time) Vegetation Optical Depth Climate Archive (VODCA). Due to slower saturation effects than and other claimed advantages over optically based

indices, the three VOD datasets will surely benefit the science community in the field of global and regional vegetation changes.

\*\*\*\*\*\*\*\*\*\*\*\*\*\*\*\*\*\*\*\*\*\*\*\*\*\*

(1) It is interesting to find that Figure 11 shows an increase of globally-averaged LAI-VOD temporal correlation in the order of C-, X-, and Ku-band. Is this pattern related to the penetration depth of microwave bands, i.e., Ku-band contains more information on top-layer leaves which are captured by LAI? While Figure 6 shows a decrease of globally-averaged LAI-VOD spatial correlation in the same order. Is this pattern related to the relatively homogenous C-band penetration depth at global scale?

(2) Instead of jointly retrieving VOD and SSM, is it possible to retrieve VOD using other sources of SSM data, e.g., GLDAS, SMOS and SMAP as inputs? An increase in SSM data quality/consistency likely improves the retrieval of VOD.

(3) It is a common practice to reduce random errors (noises) by averaging multi-sensor concurrent data. Is it your plan to incorporate more microwave sensors in the future versions of VODCA? Such sensors can include, e.g., FY-3B (X band from 2011-07-12 to present) and FY-3C (X band from 2014-05-29 to present).

(4) Do the anisotropic effects of vegetation absorption/emission play a role in VOD retrieval? That is, to what extent the cosine mapping function (in eq. 2) applies in the $0.25°$ grid, because this function is derived for horizontally homogeneous canopy. This assumption is generally unsatisfied within the $0.25°$ grid on the earth surface. Thus, the accuracy of VOD may differ with incidence angle. The MODIS maximum-value compositing NDVI (and then LAI), however, is inclined to select near-nadir pixels. In this sense, the temporal consistency is well maintained. I wonder if LAI-VOD temporal correlation is also affected by land surface heterogeneity which decreases the accuracy of VOD.

\*\*\*\*\*\*\*\*\*\*\*\*\*\*\*\*\*\*\*\*\*\*\*\*\*\*

Minor comments

Page 1, Line 19; full stop missed

Page 1, Line 23; delete redundant 'parts'

Page 2, Line 9; 'Vegetation optical depth' should be 'Vegetation Optical Depth'

Page 2, Line 18; 'analyzes' should be 'analyses'

Page 3, Line 18; 'Distribtion' should be 'Distribution'

Page 4, Line 27; 'AQUA' should be 'Aqua'

Page 5, Line 11; add a space between 'VOD' and 'from'

Page 5, Line 12; in . . . orbits

Page 6, Table 1; add the unit of GHz

Page 6, Line 7; full stop missed

Page 6, Line 9; 'us' should be 'use'

Page 9, Line 5; 'cdf' should be 'CDF'

Page 11, Line 17; 'is' should be 'are'

Page 13, Line 1; full stop missed

Page 14, Figure 7; full stop missed in the caption

Page 17, Line 10; delete 'is'

Page 18, Line 9; 'Figure 11' should be 'Figure 12'

Page 19, Line 2; please explain the meaning of number1|number2? Does 1|1 mean a contradicting trend?

Page 21, Line 14; 'implicit' should be 'implicitly'

Page 21, Line 19; the three merged VOD datasets are downstream reprocessed satellite products, and it may be inappropriate to be called 'observations'.

Last, use 'band' or 'frequency' throughout the manuscript.

---

## Referee Comment (RC1) · Anonymous Referee #1 · 28 May 2019

This manuscript presents the VOD Climate Archive (VODCA), a set of combined VOD records from several spaceborne sensors that are statistically matched to each other to cover a longer period. As such, the VODCA is very similar to the statistically matched dataset from Liu et al. (2011), as acknowledged in the paper. The VODCA differs from the Liu et al dataset in only two ways: a) it uses a slightly different form of cdf-matching to bias-correct records from different sensors and b) different versions of the data are produced for Ku-band, X-band, and C-band. Another key improvement over the Liu et al (2011) dataset, not mentioned in the paper is that this new VODCA dataset is publicly available, which should greatly increase its utility to the scientific community. While I applaud that this group is sharing this data, and while statistically matched, long records of VOD can have many applications, the authors need to do more to clarify how

this dataset is different from the Liu et al dataset. Aside from this, I have several major concerns with the underlying dataset. My specific concerns are detailed below.

Major concerns:

1) The use of regression-based cdf-matching is argued to be an important component of accurate bias-correcting. This may well be, but as currently written the paper is not convincing. The analysis in Figure 2 is not clear on this front without a label to the color map. Please include a scale bar on this figure. Even if the numbers are normalized, how much of a difference does it make? 0.5% 1%? 10% 50%? Also, it would be useful to include an additional figure that shows the exact difference between the: piece-wise CDF-matching and least-squares methods, either by replacing one of the panels or adding it as a third panel. More important, though, is the fact that no comparison between the dataset of Liu et al and this new dataset is created. In the actual practice of the VOD dataset, how closely related are the two datasets? Are there any changes induced to say, the trends? What about other statistics, or simply some sense of say, how often the VOD differs by more than some small threshold (0.05 or so) as a result of this change? Such information needs to be included in several figures and is crucial not only to judge the improvement created by this new dataset, but also towards understanding the quality of the large number of papers that have been written analyzing the Liu et al (2011) dataset.

2) Figure 7/text on page 14: The values mentioned in the text here are pretty low Spearman correlations so it is difficult to test if the LAI anomalies line up with the VOD anomalies or not. Furthermore, this analysis in and of itself does not indicate successful bias removal. It would be more useful to focus instead on whether there are any changes at the breakpoints in when different datasets are available (which are known a priori) rather than comparing across the entire record. Please use the methods developed for soil moisture in Su et al, Geophysical Research Letters, 2016 ("Homogeneity of a global multisatellite soil moisture climate data record") to test for breakpoints.

[Figure]

3) I strongly urge the authors to reconsider the choice not to include daytime retrievals in the VODCA (page 4, line 20). While the idea that daytime retrievals are more error-prone because of greater differences between soil and canopy temperature is common in the microwave radiometry community, few studies have been done document the extent of this error. Recent results suggest, for example, that PM soil moisture retrievals are not always more error-prone than AM ones (particularly under densely vegetated conditions), see Fan et al, Remote Sensing, 2015 ("The Impact of Local Acquisition Time on the Accuracy of Microwave Surface Soil Moisture Retrievals over the Contiguous United States"). Given the significant potential for diurnal changes in VOD to be useful for studying vegetation water stress (see Konings and Gentine, Global Change Biology, 2017 ("Global variations in ecosystem-scale isohydricity"), such a dataset could be quite useful. A flag could still be included for the nighttime data to suggest greater uncertainty.

4) Relatedly (page 7, line 14), the above-cited Konings and Gentine paper has shown there is a significant expected diurnal cycle in VOD (see also Konings et al, Geophysical Research Letters, 2017 "Active microwave observations of diurnal and seasonal variations of canopy water content across the humid African tropical forests" for the active equivalent of this, with more complete diurnal measurements). As such, presenting the data to be "resampled to a specific time" as in page 7, line 14, is misleading. At the very least, the data should be presented as averaged over a certain period. If that is the case, and if the authors really insist on not using daytime data, it would still be cleaner to just present it as a day-long average.

5) Little literature exists on whether sensor differences are really more significant than algorithmic differences. This may explain why SMOS and SMAP baseline retrievals were found by the authors to have little consistency; those retrieval algorithms are fairly different – not something specific about L-band frequency. L-band VOD has been shown to have significant utility over X-band and likely C-band (see Brandt et al, Nature Ecology and Evolution, 2018 "Satellite passive microwaves reveal recent climateinduced carbon losses in African drylands"). While I recognize that it may genuinely be impossible to create an L-band product using the methodology employed here (without re-running the LPRM on SMOS and SMAP so that a common algorithm is present), a more detailed treatment should be provided than just lines 21-23 on page 3. Please make it clear that it is not possible for you to create the L-band product, not "does not warrant a product for it", which suggests L-band data is not useful. Also, can you include the low temporal correlation in a supplemental plot?

Other presentational issues:

1) Section 3.1 Regarding the 2 AMSR2 C-band channels: was any statistical evaluation done to see how different the retrievals from the two channels were, when taken in isolation?

2) Page 9 lines 2-3: Please provide more information on how the bin sizes are chosen

3) Page 10 line 11: How often does removal of such unphysical values happen? This is important information as these values are made unphysical as a direct consequence of the cdf-matching variant employed here.

4) Page 10, line 1-2: How many observations is deemed enough? Please include in text (not just Figure 3 for clarity. Relatedly, Figure 3 is not consistent with the text in Section 3.2.4 (since for example, the figure pseudo-code does not mention the use of the first and last two years). This makes it actively confusing – please make sure Figure 3 is fully complete

Minor textual issues:

The abstract would be cleaner if it was all one paragraph

Page 2 line 9: please fix the capitalization of how VOD is written out

Page 6 line 9 "us" instead of "use"

Page 7 line 4: please reword

Page 7 line 22: please include a comma

Figure 4b: How are different VOD datasets combined in this figure if this is "the original VOD data" as stated in the legend?

Page 12, line 2: Please define TC, SV, and BG

Page 17, line 10: Contributing

Page 19, lines 29-31: Is a flag included to warn the user of this?

---

## Referee Comment (RC2) · M. Piles (Referee) · 14 Jun 2019

General comments

The authors present alongside this paper a new long-term data set of vegetation optical depth (VOD) at C, X and Ku bands constructed based on the statistical merging of available products spanning the last three decades. They first describe the motivation and their proposed approach, and then detail some aspects of the data set such as its spatio-temporal coverage and its error characteristics with respect to the individual products. Since direct validation of VOD data is not possible due to the lack of in situ measurements, they provide an indirect validation by means of a spatio-temporal comparison to two optically-derived measurements: Leaf Area Index and Vegetation

[Figure]

Continuous Fields data.

I would like to first thank the authors for their initiative of creating a new data set of multi-frequency long-term VOD and making it freely accessible. I read the manuscript with interest. The presented data set has enormous potential to monitor global changes in canopy water and contribute to a wide variety of ecological studies. However, I found some major aspects need to be further detailed or analyzed in the manuscript to support the usability and robustness of the product. My main concern is that the manuscript does not contain a full characterization of their matching methodology and, as a consequence, it is not clear how the flags provided correspond to the quality of the retrievals and how the final spatio-temporal resolution of the products is impacted. This is key information for potential users. Also, some design criteria need to be further justified or discussed in the text and their validation approach, while convincing, does not show the value of having multi-frequency VOD. The novelty and potential applications of the provided data set need to be further supported by references to previous related works.

My main recommendations follow:

1. Their approach for the merging builds from the one used for the ESA CCI Soil Moisture product and the previous long-term VOD product from Liu et al., 2011, with improvements to make it more robust to the presence of outliers. The improvements shown with respect to the previous version is not convincing. What is the numerical range of the colorbar in Fig. 2? Can the authors also show results with real data? Also, the authors say (page 8, line 31) they dynamically increase the step size of the percentiles "if only a few" observations are available. It would be important to be more specific here and show how the method is sensitive to the choice of this parameter. In general, an improved characterization of their matching approach is needed.

2. The authors report there is a flag indicating the matching method (page 10, line 8) and a flag indicating which sensors contributed to a measurement (page 11, line

2). It would be very useful if they could relate those flags to the quality of the final product and make recommendations to the user. Perhaps the authors could consider dedicating a specific section of the paper to their quality flags and assessment.

3. I would strongly recommend the authors to consider including the daytime observations to the data set. Although it is well-known that daytime retrievals are expected to have a higher error than nighttime ones due to the thermal equilibrium assumed in the inversion, the difference between day and night canopy water have been shown useful for certain science studies (e.g. see Konings & Gentine, "Global variations in ecosystem‐scale isohydricity", Global Change Biology, 2016). Also, their combination could be potentially useful for some applications to enhance the temporal coverage.

4. The validation does not show the value of the multi-frequency retrievals, nor discusses in detail their differences with respect to the optical indicators they selected. The authors should elaborate more on their results with focus on the different bands and perhaps consider a comparison of the sensitivity of the different VOD to biomass (e.g. see Nemesio-Rodríguez et al., biogeosciences, 2018).

Specific comments and recommendations:

1. Page 1, line 10. The authors should introduce in the abstract the previous long-term VOD data set and clarify the novelties of their newly presented data set, i.e. frequency-specific VOD, extended period, improved matching.

2. Page 1, line 24. Is the trend measured by all frequencies? Are there any differences? It would be nice to complement the validation and include the value of having frequency-specific VOD here.

3. Page 2, line 2. The authors could (at least) indicate how the multi-frequency VOD could actually complement optical measurements (e.g. canopy water vs. greenness)

4. Page 2, line 14. Additional references are needed in the intro and the discussion regarding multi-frequency VOD estimates and sensitivity to different parts of the canopy. I point out two articles hereafter, but recommend nonetheless the authors to do a bibliography search:

F. Tian et al., Coupling of ecosystem-scale plant water storage and leaf phenology observed by satellite, nature ecology and evolution, 2018

N. Rodríguez-Fernández et al., An evaluation of SMOS L-band vegetation optical depth (L-VOD) data sets: high sensitivity of L-VOD to above-ground biomass in Africa, biogeosciences, 2018

5. Page 3, line 23. Do the authors mean there is a low temporal correlation of SMAP and SMOS VOD products? Which products? Please, provide appropriate references or supporting material for this statement. Perhaps the addition of L-band could be directly included as future work, latest products from the two missions (SMAP MTDCA and SMOS-IC for instance) seem to agree well.

6. Page 4, line 10. A reference to the tau-omega model is needed. Please include: T. Mo, B. Choudhury, T. Schmugge, and T. Jackson, "A model for microwave emission from vegetation-covered fields," J. Hydrol., vol. 184, no. C13, pp. 101–129, Dec. 1982.

7. Table 1: It would be interesting to add ascending and descending times for each sensor as well as their incidence angles, spatial and temporal resolutions. The authors could perhaps add a little discussion on the impacts of mixing the different times and observation geometries (spatial resolution, incidence angle, etc).

8. Page 4, line 24: it is unclear how the different data sets can be accessed (webpage?). Please, specify which ones are available and which ones are not (perhaps on Table 2 also).

9. Page 9, Line 25. I understand AMSR-E is used as a reference for having the highest overlap. But perhaps AMSR-2 could also be chosen for being a more advanced instrument with improved capabilities, or also a modeled VOD could potentially be used.

[Figure]

Please, include a discussion for this choice (or provide a reference) and why it was chosen over the alternatives.

10. Page 6, line 1. Please, indicate how to access the ancillary data used in the corresponding subsection (LAI and VCF).

11. Page 6, line 4. What do the authors expect from the comparison of VOD and LAI? A rationale of why they chose LAI over other indices (e.g. NDVI, EVI) and whether they expect a higher correlation with any of the specific VOD products is needed.

12. Page 7, line 8: How is the VOD climatology from AMSR-E derived? Please provide details.

13. Page 7, line 26. I agree with the authors that negative VOD retrievals are physically impossible. However, they are most probably linked to uncertainties/simplifications on the physical model used in the inversion, and their direct truncation may lead to erroneous trends for specific areas. One alternative could be to let the user truncate the values after temporally averaging the data set according to the needs of their study. This is the procedure followed for instance in the SMOS-IC product (Fernández-Moran et al., remote sensing, 2017). I would ask the authors to consider this option or at least, mention it in the discussion.

14. Page 8, line 2. How different are the retrievals from the two C-band channels? Again the authors include a flag but this flag is not useful if it is not related to a quality indicator or any further recommendation is given.

15. Page 8, line 22. I infer from the text that there is a need to a new cdf-matching technique due to the presence of outliers in the VOD data set. Could this cdf-matching also improve the VOD data in Liu et al 2009, 2011? Could this cdf-matching improve the soil moisture merging within ESA CCI? The authors could perhaps elaborate on this, to better motivate the approach.

16. Page 10, line 11. Does this happen very often? Could this be one aspect to

improve to increase coverage?

17. Page 10, line 18. Have the authors tried with the median statistic? It is less sensitive to ouliers.

18. Figure 3. I do not think this figure is necessary.

19. Figure 4. What is the dominant vegetation in the chosen pixel? Perhaps the author could also include an example of time series in which TMI is also used, for completeness.

20. Page 12, line 3. The authors could also perhaps refer to the L-band VOD spatial patterns, which are consistent and correlate well with canopy height (e.g. Konings et al., L-band vegetation optical depth and effective scattering albedo estimation from SMAP, Remote Sensing of Environment, 2017).

21. Figure 6. It is hard to see the seasonal patterns. The authors could perhaps consider showing only the period 2002-2017 (or even shorter)

22. Page 15, line 1. It is unclear how the authors measure the spatio-temporal coverage. Is Fig.8 showing the fraction of days each month as stated in the label? The final temporal resolution shown in the figure and referenced in the text above is unclear.

23. Page 15, line 14. What do the authors understand by a "CDF-matching failure"? could it be for one specific reason (e.g. see comment #15 above), or several? please, be more specific.

24. Page 19, line 29. This advice is helpful but it could really be useful and applicable if converted into a criteria that contributes to a quality flag. There is clearly a need for a quality flag.

25. Fig. 12. I would suggest to include subfigures f and g into a separate figure, for clarity.

26. Page 21, line 17. Only Ku band spans three decades, this sentence is a bit

misleading.

27. Page 21, line 21. From section 4.2., it is unclear that the resulting VOD data sets provide observations "on a daily basis".

28. Page 21, line 24. The authors could perhaps consider mentioning at some point in the manuscript that their work is particularly relevant in the context of the prospect launch of the multi-frequency candidate mission Copernicus Microwave Imaging Radiometer (CIMR, www.cimr.eu).

A few technical corrections:

Page 6, line 9: should be "Use"

Page 6, line 11: typo "short vegetation all shorter vegetation"

Page 7, line 4: typo "for the product of each product"

Page 7, lin 21. Reference to Nijs et al (2015) seems to be missing.

Page 17, line 10: typo "is still positively contribute"

---

## Author Comment (AC1) · 19 Jul 2019

**Author comments to: The Global Long-term Microwave Vegetation Optical Depth Climate Archive VODCA**

Leander Moesinger[1], Wouter Dorigo[1], Richard de Jeu[2], Robin van der Schalie[2], Tracy Scanlon[1], Irene Teubner[1], and Matthias Forkel[1]

[1]Vienna University of Technology, Department of Geodesy and Geoinformation, Gußhausstraße 27-29, 1040 Vienna, Austria
[2]VanderSat, Wilhelminastraat 43A, 2011 VK Haarlem, The Netherlands

**Correspondence:** Leander Moesinger (Leander.Moesinger@geo.tuwien.ac.at, vodca@geo.tuwien.ac.at)

Formatting as follows:

Reviewers' comments

Reply to comments

[page:line]

5   [page:line] Added/changed parts to the manuscript

**1   Response to Xingwang Fan**

(1) It is interesting to find that Figure 11 shows an increase of globally-averaged LAI-VOD temporal correlation in the order of C-, X-, and Ku-band. Is this pattern related to the penetration depth of microwave bands, i.e., Ku-band contains more informa-
10   tion on top-layer leaves which are captured by LAI? While Figure 6 shows a decrease of globally-averaged LAI-VOD spatial correlation in the same order. Is this pattern related to the relatively homogeneous C-band penetration depth at global scale?

This is very intriguing, we did not realize this before. First thing to note is that for Figure 11 for each location the Spearman correlation between the LAI and VOD time series is calculated and then averaged globally, while for Figure 6 directly
15   the correlation between the hovmoeller diagrams is determined. As such Figure 11 represents the globally averaged temporal correlation, while Figure 6 is related to both spatial AND temporal correlation.
As such we think that the order of correlations in fig 11 is a result of Ku-band and LAI being more affected by the top vegetation canopy than the other bands. However, Figure 6 is a more complicated matter. It could also be that the differences in correlation are due to the differing spatial extent. For example, C-band has more spatial gaps due to more RFI being present
20   than in the other bands. The C- and X-band coverage agrees better with the LAI coverage than Ku-band since all points of a latitude are averaged. Therefore, we should not over-interpret these coefficients. As such we will remove them from the paper to avoid confusion or wrong conclusions.

[12 : 7]

[14 : 1]

(2) Instead of jointly retrieving VOD and SSM, is it possible to retrieve VOD using other sources of SSM data, e.g., GLDAS, SMOS and SMAP as inputs? An increase in SSM data quality/consistency likely improves the retrieval of VOD.

Radiative Transfer Model inversion approaches like LPRM aim to retrieve VOD and SSM in a consistent manner that guarantees energy conservation laws. When forcing the retrieval with external soil moisture data sets you also force the VOD to fit the observed brightness temperatures, which is certainly not a guarantee for a good VOD retrieval. LPRM is a proven and tested methodology for retrieving both SSM and VOD simultaneously. For example, using modeled data from a reanalyis data set like GLDAS-Noah or ERA5 would also introduce errors (e.g. related to errors in precipitation forcing) that would directly translate back into the skill of the VOD retrieval. SSM derived from SMAP or SMOS has other uncertainties. In summary, to guarantee consistency and independency of SSM and VOD retrievals, no external SSM data set is used in the retrieval of VOD.

(3) It is a common practice to reduce random errors (noise) by averaging multi-sensor concurrent data. Is it your plan to incorporate more microwave sensors in the future versions of VODCA? Such sensors can include, e.g., FY-3B (X band from 2011-07-12 to present) and FY-3C (X band from 2014-05-29 to present).

We are definitely looking to include as many different sensors as possible to further reduce random errors. But we were not able to get access to Fen Yung data or WindSat data past 2012, even though we would very much like to include it. More realistically, future VODCA versions will include GPM-based retrievals.

(4) Do the anisotropic effects of vegetation absorption/emission play a role in VOD retrieval? That is, to what extent the cosine mapping function (in eq. 2) applies in the 0.25 grid, because this function is derived for horizontally homogeneous canopy. This assumption is generally unsatisfied within the 0.25 grid on the earth surface. Thus, the accuracy of VOD may differ with incidence angle. The MODIS maximum-value compositing NDVI (and then LAI), however, is inclined to select near-nadir pixels. In this sense, the temporal consistency is well maintained. I wonder if LAI-VOD temporal correlation is also affected by land surface heterogeneity which decreases the accuracy of VOD.

Anisotropy primarily plays a role in observations of reflected radiance. This is the case e.g. for solar-reflective observations (bi-directional reflectance distribution function) like those used for MODIS LAI retrievals, or from active microwave observations, which are affected by canopy structure. Emitted radiance can generally be considered anisotrope. For almost all of the currently used soil moisture retrieval algorithms it is an accepted and applied assumption to have polarization-independent VOD. H and V-polarizations do not result in differences in VOD retrievals (Owe et al., 2001) and is actually the basis of LPRM.

The purpose of the maximum-value compositing technique for NDVI data aims to reduce the effects of off-nadir observations and to reduce the effect of low NDVI values on the vegetation signal that are for example caused by snow cover or atmospheric distortions. The maximum NDVI value within a period hence helps to extract a vegetation-sensitive NDVI and to reduce other effects. As VOD retrievals are not affected by snow cover or atmospheric distortions, there is no need to apply the maximum value composite technique to VOD time series (unless there is a biogeophysical interest in maximum VOD values).

**2 Response to reviewer #1**

**2.1 Main discussion points**

1) The use of regression-based cdf-matching is argued to be an important component of accurate bias-correcting. This may well be, but as currently written the paper is not convincing. The analysis in Figure 2 is not clear on this front without a label to the color map. Please include a scale bar on this Figure. Even if the numbers are normalized, how much of a difference does it make? 0.5% 1%? 10% 50%? Also, it would be useful to include an additional Figure that shows the exact difference between the: piece-wise CDF-matching and least-squares methods, either by replacing one of the panels or adding it as a third panel. More important, though, is the fact that no comparison between the data set of Liu et al and this new data set is created. In the actual practice of the VOD data set, how closely related are the two data sets? Are there any changes induced to say, the trends? What about other statistics, or simply some sense of say, how often the VOD differs by more than some small threshold (0.05 or so) as a result of this change? Such information needs to be included in several Figures and is crucial not only to judge the improvement created by this new data set, but also towards understanding the quality of the large number of papers that have been written analyzing the Liu et al (2011) data set.

- We did not include a scale bar to the plot because it is a synthetic experiment and as such the absolute values are a function of the parameters (e.g. the distribution the values are sampled from) and therefore not very useful. But we agree that labeling the scale bar would help to get a feeling for the magnitude of the values, see Fig. 1 for new version with a normalized scale bar.

- In response to the comment that it would be useful to include an additional figure that shows the exact difference between the piece-wise CDF-matching and least-squares methods, we tried to make a illustrative figure showing the differences between the two methods. Unfortunately, it ended up being more confusing than helpful. For this reason we suggest to use figure 1 (with the modified scale bar) to show that only the first and last percentile bins are affected.

- Comparing VODCA to Liu et al's data set is a good idea. In the revised manuscript, for several analyses we will compare the finished VODCA products to Liu et al's data set. In summary, we found that the most important difference is that VODCA consists of independent daily observations while the Liu data set seems to have some smoothing applied to the

original observations. This becomes evident for days where no microwave observations are available: In Liu's data set such short-term gaps are filled, although we could not find in literature how this was exactly done.

We see more or less the same trends in VODCA-Ku as in the Liu et al. data set (Fig. 2). Using daily data, the Liu data set correlates more strongly with LAI than VODCA (Fig. 3, left column), probably because of the applied temporal smoothing. But if both data sets are downsampled to monthly time steps, VODCA correlates more strongly with LAI than Liu et al's data set (Fig. 3, right column). Based on our analyses we conclude that papers written on the basis of Liu et al's data set are still valid, but that VODCA will add value to future studies because it covers a longer time span, has temporally-independent daily VOD values, provides separate products for different frequencies, and reduced noise.

We will include the comparison of the VODCA data set with Liu's data. The main changes to the manuscript will be:

– Page 17, subsection "4.4.1 Correlation between VOD and LAI" now also compares Liu et al's data set with LAI.

– Updated most hovmoeller (see fig. 4) plots to also include Liu et al's data set. We will not update the fractional coverage hovmoeller, as the smoothing present in Liu et als data set leads to inflated values.

– Added figure 2 to section "4.4.2 Trend—analysis of VOD, LAI and Vegetation Continuous Fields" with description.

2) Figure 7 on page 14: The values mentioned in the text here are pretty low Spearman correlations so it is difficult to test if the LAI anomalies line up with the VOD anomalies or not. Furthermore, this analysis in and of itself does not indicate successful bias removal. It would be more useful to focus instead on whether there are any changes at the breakpoints in when different data sets are available (which are known a priori) rather than comparing across the entire record. Please use the methods developed for soil moisture in Su et al, Geophysical Research Letters, 2016 ("Homogeneity of a global multi-satellite soil moisture climate data record") to test for breakpoints.

We agree with the reviewer that computing the correlations per "blending period" gives more insight in the skill to detect anomalies per period. Due to the absence of a VOD reference data set, the Su et al methods cannot be applied. Instead, we calculated the correlation of VODCA and LAI for the different blending period, similar to Dorigo et al. (2015). The results (Fig. 5) show that the spatial distribution of the correlation between VOD and LAI is time-invariant for all VODCA bands. This demonstrates that the temporal dynamics are consistent over the whole time period.

Correlation analysis between LAI and VOD per blending period (fig. 5) will be added to the results. As MODIS LAI only starts in 2000, the blending periods are:

– 00-02 (SSMI + TMI)

– 02-09 (SSMI + TMI + AMSRE + WindSat)

– 09-12 (TMI + AMSRE + WindSat)

[Figure]

**Figure 1.** Reworked figure 2: Simulated variance of slopes of old a new CDF matching method. New are the scale bar tick labels

[Figure]

(a) Liu et al trends

(b) VODCA-Ku trends

**Figure 2.** Liu et al and VODCA trends from 1993-01 to 2012-12 (will be included in revised manuscript).

[Figure]

**Figure 3.** Correlation of the Liu et al. and the VODCA products with MODIS LAI for daily (left) and monthly (right) observations. For the daily analysis, only dates where both all four data sets have values are used. For the monthly analysis, the data are first resampled to monthly averages, afterwards only months where all four data sets have values are used.

[Figure]

(a) Mean

(b) Anomalies

**Figure 4.** Hovmoellers diagrams of monthly VOD values (top) and monthly VOD anomalies (bottom) including the merged VOD dataset by Liu et al.

[Figure]

**Figure 5.** Correlation between VODCA and MODIS LAI for raw time series and seasonal anomalies for all bands and different blending periods. The blending periods are based on Fig. 1 in the manuscript. VODCA-X has no global coverage between 2000 to 2002 and therefore this period is not included.

– 12-15 (TMI + AMSR-2)

– 15-19 (AMSR-2)

   3) I strongly urge the authors to reconsider the choice not to include daytime retrievals in the VODCA (page 4, line 20).
125 While the idea that daytime retrievals are more error-prone because of greater differences between soil and canopy temperature is common in the microwave radiometry community, few studies have been done document the extent of this error. Recent results suggest, for example, that PM soil moisture retrievals are not always more error-prone than AM ones (particularly under

densely vegetated conditions), see Fan et al, Remote Sensing, 2015 ("The Impact of Local Acquisition Time on the Accuracy of Microwave Surface Soil Moisture Retrievals over the Contiguous United States"). Given the significant potential for diurnal changes in VOD to be useful for studying vegetation water stress (see Konings and Gentine, Global Change Biology, 2017 ("Global variations in ecosystem-scale isohydricity"), such a data set could be quite useful. A flag could still be included for the nighttime data to suggest greater uncertainty.

While technically it would be possible to produce a daytime product using the same methods, the daytime LPRM-VOD products are still very experimental. Currently, we don't want to release a daytime product to prevent users from making false scientific conclusions based on potential data artifacts. Our experience from ESA CCI Soil Moisture has taught us, that despite providing quality flags and extensive documentation, many users do make wrong use of data sets. A release of daytime products requires a proper evaluation, a comparison with nighttime products and an assessment of differences. Such an analysis is beyond the scope of this paper. However, we consider such an analysis essential and will likely address it in the near future. Once our scientific understanding and confidence in the day-time products is mature enough, we will include this in a future release of VODCA.

4) Relatedly (page 7, line 14), the above-cited Konings and Gentine paper has shown there is a significant expected diurnal cycle in VOD (see also Konings et al, Geophysical Research Letters, 2017 "Active microwave observations of diurnal and seasonal variations of canopy water content across the humid African tropical forests" for the active equivalent of this, with more complete diurnal measurements). As such, presenting the data to be "resampled to a specific time" as in page 7, line 14, is misleading. At the very least, the data should be presented as averaged over a certain period. If that is the case, and if the authors really insist on not using daytime data, it would still be cleaner to just present it as a day-long average.

We concur, this is a bit poorly worded and can be misunderstood.
We are adjusting the relevant passages of the manuscript to talk of "nightly averages" instead.

5) Little literature exists on whether sensor differences are really more significant than algorithmic differences. This may explain why SMOS and SMAP baseline retrievals were found by the authors to have little consistency; those retrieval algorithms are fairly different – not something specific about L-band frequency. L-band VOD has been shown to have significant utility over X-band and likely C-band (see Brandt et al, Na-ture Ecology and Evolution, 2018 "Satellite passive microwaves reveal recent climate-induced carbon losses in African drylands"). While I recognize that it may genuinely be impossible to create an L-band product using the methodology employed here (without re-running the LPRM on SMOS and SMAP so that a common algorithm is present), a more detailed treatment should be provided than just lines 21-23 on page 3. Please make it clear that it is not possible for you to create the L-band product, not "does not warrant a product for it", which suggests L-band data is not useful. Also, can you include the low temporal correlation in a supplemental plot?

[Figure]

(a) L-band LPRM-SMAP vs. LPRM-SMOS          (b) Ku-band AMSR-E vs. WindSat

**Figure 6.** Correlations between different sensors of the same band. The Ku-band WindSat vs. AMSR-E plot is similar to all other sensor combinations in the Ku, X, and C band.

We agree that a more sophisticated argumentation would be appropriate at this point. Also Reviewer #2 was pointing this out. In a preliminary analysis, we used L-band VOD products from SMAP and SMOS retrieved with LPRM. The temporal correlation between the daily LPRM-SMOS and LPRM-SMAP values is very low (globally in average about 0.1, while the correlation coefficients in the other bands achieved values of 0.6 to 0.7, Fig. 6). Lower temporal dynamics and hence correlations are expected for L-band in comparison to shorter wavelengths because L-band largely penetrates the canopy with strong seasonal changes in leaf biomass and is more sensitive to the woody parts. Hence, the relatively small intra-annual dynamics are more sensitive to noise in the data. This is not a problem exclusive to LPRM-derived L-band VOD products. To the best of our knowledge, all studies involving L-band VOD use temporally averaged data rather than using daily values. For example, Brandt et al. (2018) averaged all SMOS-IC data between 2010 and 2016 and analyzed only spatial correlations, disregarding temporal dynamics.

We also applied the VODCA merging procedure to L-band VOD data from LPRM-SMAP and LPRM-SMOS. The autocorrelation analysis showed that the obtained VODCA-L-band VOD has a lower temporal autocorrelation than the original LPRM-SMAP VOD (Fig. 7). This indicates that the level of noise in L-band was increased with the merging. Hence for L-band, the merging results in a lower-quality data set. In addition, the low density of observations in LPRM-SMOS causes a highly unbalanced temporal coverage of VOD values (Fig. 8). Given this unbalanced data coverage there is a high risk that users might wrongly use this dataset for e.g. trend analysis. However, we are convinced that there is a large potential to produce a more reliable multi-sensor/multi-product merged L-band data set in the future, e.g. by using alternative L-band retrievals from SMOS-IC or MTDCA. Yet, the current VODCA methodology cannot be easily applied to L-band data. Hence alternative blending approaches first need to be thoroughly assessed, which goes beyond the scope of the current paper.

We will add a paragraph in the introduction explaining in detail the reasoning behind not producing VODCA-L. We will also include figures 6 and 7 in the supplement for that purpose and expand the discussion regarding the possibility of an L-band product. We will also take care to talk specifically of LPRM-L-VOD and not L-VOD in general.

[Figure]

(a) SMOS change                               (b) SMAP change

**Figure 7.** Ar(1) change of SMAP and SMOS due to merging. Note that the original SMAP time series has less noise than experimental VODCA-L

**2.2 Minor discussion points**

1) Section 3.1 Regarding the 2 AMSR2 C-band channels: was any statistical evaluation done to see how different the retrievals from the two channels were, when taken in isolation?

190

VOD from the two C-band channels from AMSR2 is highly correlated (Fig. 9), so using 7.3 GHz instead of 6.9 Ghz will have little impact. Actually, the 7.3 GHz channel was purposely added to AMSR2 to mitigate RFI in the 6.9 GHz channel. We will add this plot to the supplement and add to the main text:

195     [8 : 3] As the two C-bands are highly correlated, the use of one or the other has a minor impact on the quality of the dataset (Fig. 9).

2) Page 9 lines 2-3: Please provide more information on how the bin sizes are chosen.

200     Agreed, this is in need of some clarification. We added following sentences to the end of the relevant paragraph:
[9 : 2] The bin size of 20 was chosen as compromise between data coverage and often used bin sizes. A bin size of 50 observations is often used as a rule of thumb for univariate regression to get robust estimates (Green, 1991). However, our main goal was rather to prevent time series with very few observations from learning spurious scaling parameters and we also did not want to loose all time series with less than 50 values. As such 20 was chosen as a compromise.

205

3) Page 10 line 11: How often does removal of such unphysical values happen? This is important information as these values are made unphysical as a direct consequence of the cdf-matching variant employed here.

A fraction in the order of $1/10^6$ to $1/10^8$ of values are lost this way, so almost nothing gets lost. Anyway, the only way to 210     prevent any CDF-matching technique to produce values below zero is to force its intercept trough zero. But this would mean

[Figure]

**Figure 8.** Fractional coverage hovmoeller plot, also including experimental VODCA-L.

that potentially it becomes a very bad fit for the value range where the data is actually located. We added the clarification to the manuscript.

[10 : 11] These values are deemed unreliable and are removed. However, this occurs very rarely, only a fraction of about $1/10^6$ to $1/10^8$ of values are lost in this way.

4) Page 10, line 1-2: How many observations is deemed enough? Please include in text (not just Figure 3 for clarity. Relatedly, Figure 3 is not consistent with the text in Section 3.2.4 (since for example, the figure pseudo-code does not mention the

[Figure]

**Figure 9.** Correlation between the AMSR2 6.9 and 7.3 GHZ band

use of the first and last two years). This makes it actively confusing – please make sure Figure 3 is fully complete.

220

As a threshold, we used the same number of observations as for the bins reported on Page 9 lines 2-3, i.e. 20 (otherwise the matching may become too unreliable).

About Fig. 3: The usage of the first/last 2 years is mentioned in the pseudo-code but we agree that it does not mention that
225   the minimum of 20 observations need to come from the last two years of AMSR-E and the first two years of AMSR2. But anyway, since reviewer M. Piles finds the Figure unnecessary and we also were never content with it, we will remove Figure 3 entirely without replacement.

Figure 3 will be removed from the manuscript.

230

**3 Response to Maria Piles**

**3.1 Main discussion points**

1. Their approach for the merging builds from the one used for the ESA CCI Soil Moisture product and the previous long-term VOD product from Liu et al., 2011, with improvements to make it more robust to the presence of outliers. The improvement

shown with respect to the previous version is not convincing. What is the numerical range of the colorbar in Fig. 2? Can the authors also show results with real data? Also, the authors say (page 8, line 31) they dynamically increase the step size of the percentiles "if only a few" observations are available. It would be important to be more specific here and show how the method is sensitive to the choice of this parameter. In general, an improved characterization of their matching approach is needed.

We agree that Figure 2 does not give a full insight into the improvements of the methods. However, with the lack of any reference data, making some test with real data is unfeasible. Using synthetic data has the advantage that the results are not skewed by some external effect, but only depend on the methods used. These simulations show that for exactly the same set of input data, the new CDF-matching method gives much more reliable scaling parameters for the first and last bins. We will add a normalized scale bar for reference to quantify the change. A bin size of 20 was chosen as compromise between data coverage and often used bin sizes. A bin size of 50 observations is often used as a rule of thumb for uni-variate regression to get robust estimates (Green, 1991). However, our main goal was rather to prevent time series with very few observations from learning spurious scaling parameters and we also did not want to loose all time series with less than 50 values. As such 20 was chosen as a compromise.

[9 : 2] The bin size of 20 was chosen as compromise between data coverage and often used bin sizes. A bin size of 50 observations is often used as a rule of thumb for uni-variate regression to get robust estimates (Green, 1991). However, our main goal was rather to prevent time series with very few observations from learning spurious scaling parameters and we also did not want to loose all time series with less than 50 values. As such 20 was chosen as a compromise.

2. The authors report there is a flag indicating the matching method (page 10, line 8) and a flag indicating which sensors contributed to a measurement (page 11, line 2). It would be very useful if they could relate those flags to the quality of the final product and make recommendations to the user. Perhaps the authors could consider dedicating a specific section of the paper to their quality flags and assessment.

First of all, we want to point out that right now only the C-band can have any bad quality flags due to irregular processing, the other bands always use the standard processing chain. The 6.9 and 7.3 GHz C-bands are highly correlated (Fig. 9), showing that using the lower frequency instead of 6.9 GHz has little impact on the results. For the other flags it is really hard to make any recommendations as they depend a lot on what the data are used for and there is a wide range of possible uses that we cannot foresee. Still, we will add to the supplement a summary of all available quality flags and some possible effects of them.

Supplement: We added a section about quality flags, their meaning with links to relevant sections in main text.
In main text, following line is added together with the figure:

 As the two C-bands are highly correlated, the use of one or the other has a minor impact on the quality of the dataset (Fig. 9).

3. I would strongly recommend the authors to consider including the daytime observations to the data set. Although it is well-known that daytime retrievals are expected to have a higher error than nighttime ones due to the thermal equilibrium assumed in the inversion, the difference between day and night canopy water have been shown useful for certain science studies (e.g. see Konings & Gentine, Global variations in ecosystemscale isohydricity, Global Change Biology, 2016). Also, their combination could be potentially useful for some applications to enhance the temporal coverage.

While technically it would be possible to produce a daytime product using the same methods, the daytime LPRM-VOD products are still very experimental. Currently, we do not want to release a daytime product to prevent users from making false scientific conclusions based on potential data artifacts. Our experience from ESA CCI Soil Moisture has taught us, that despite providing quality flags and extensive documentation, many users do make wrong use of data sets. A release of daytime products requires a proper evaluation, a comparison with nighttime products and an assessment of differences. Such an analysis is beyond the scope of this paper. However, we consider such an analysis essential and will likely address it in the near future. Once our scientific understanding and confidence in the day-time products is mature enough, we will include this in a future release of VODCA.

4. The validation does not show the value of the multi-frequency retrievals, nor discusses in detail their differences with respect to the optical indicators they selected. The authors should elaborate more on their results with focus on the different bands and perhaps consider a comparison of the sensitivity of the different VOD to biomass (e.g. see Nemesio-Rodríguez et al., biogeosciences, 2018)

While we are eager to analyze the usability of the different frequencies for various applications, this is not the focus of a data set paper. The primary goal of the manuscript is to introduce the new VODCA data set, give insights into its methodology, and demonstrate in various ways if the values we produce and the dynamics that we see in the data set are plausible. This focus is also outlined in the aims and scope of the journal: https://www.earth-system-science-data.net/about/aims_and_scope.html. Focusing too much on the interpretation of the results would derail the topic and further increase the length of the paper (it already now is on the long side).

**3.2 Minor discussion points**

1. Page 1, line 10. The authors should introduce in the abstract the previous long-term VOD data set and clarify the novelties of their newly presented data set, i.e. frequency-specific VOD, extended period, improved matching.

While Liu et al's data set has a much more prominent role in the revised manuscript, the abstract becomes too long and confusing if we describe two VOD data sets and their differences already there. However, the extended period, the new matching technique, and the separate frequency-specific VOD data sets are already mentioned.

2. Page 1, line 24. Is the trend measured by all frequencies? Are there any differences? It would be nice to complement the validation and include the value of having frequency-specific VOD here.

Thanks for the suggestion, this is indeed something that needs to be specified. We only looked at Ku-band long term trends (because it is 10 years longer than X-band).

[1 : 24] We added: "Trend analysis of Ku-Band VODCA shows that between 1987 and 2017 there has been ..."

3. Page 2, line 2. The authors could (at least) indicate how the multi-frequency VOD could actually complement optical measurements (e.g. canopy water vs. greenness)

This is a good idea, we added a sentence on the uniqueness of the observations.

[3 : 2] We added: "In summary, VODCA shows vast potential for monitoring spatial-temporal ecosystem changes complementary to existing long-term vegetation products from optical remote sensing as VOD is unaffected by cloud cover or high sun zenith angles. In addition, VOD is sensitive to vegetation water content and hence complements optical indices of vegetation greenness and leaf area.

4. Page 2, line 14. Additional references are needed in the intro and the discussion regarding multi-frequency VOD estimates and sensitivity to different parts of the canopy. I point out two articles hereafter, but recommend nonetheless the authors to do a bibliography search: F. Tian et al., Coupling of ecosystem-scale plant water storage and leaf phenology observed by satellite, nature ecology and evolution, 2018N. Rodríguez-Fernández et al., An evaluation of SMOS L-band vegetation optical depth (L-VOD) data sets: high sensitivity of L-VOD to above-ground biomass in Africa, biogeosciences, 2018

Thanks for the suggestion, we added your proposed as well as additional references.

Changed sentence: Short wavelengths experience a higher attenuation by vegetation (and hence relate to higher VOD values) than longer ones (Liu et al., 2009; Owe et al., 2008; Kerr et al., 2018). As a consequence, VOD estimates from long wavelengths are sensitive to deeper vegetation layers (e.g. stem biomass) while VOD estimates from short wavelengths are more sensitive to canopy moisture content (Chaparro et al., 2018; Tian et al., 2018; Fan et al., 2018; Konings et al., 2019).

5. Page 3, line 23. Do the authors mean there is a low temporal correlation of SMAP and SMOS VOD products? Which products? Please, provide appropriate references or supporting material for this statement. Perhaps the addition of L-band could be directly included as future work, latest products from the two missions (SMAP MTDCA and SMOS-IC for instance) seem to agree well.

We agree that a more sophisticated argumentation would be appropriate at this point. Also Reviewer #1 was pointing this out. In a preliminary analysis, we used L-band VOD products from SMAP and SMOS retrieved with LPRM. The temporal correlation between the daily LPRM-SMOS and LPRM-SMAP values is very low (globally in average about 0.1, while the correlation coefficients in the other bands achieved values of 0.6 to 0.7, Fig. 6). Lower temporal dynamics and hence correlations are expected for L-band in comparison to shorter wavelengths because L-band largely penetrates the canopy with strong seasonal changes in leaf biomass and is more sensitive to the woody parts. Hence, the relatively small intra-annual dynamics are more sensitive to noise in the data. This is not a problem exclusive to LPRM-derived L-band VOD products. To the best of our knowledge, all studies involving L-band VOD use temporally averaged data rather than using daily values. For example, Brandt et al. (2018) averaged all SMOS-IC data between 2010 and 2016 and analyzed only spatial correlations, disregarding temporal dynamics.

We also applied the VODCA merging procedure to L-band VOD data from LPRM-SMAP and LPRM-SMOS. The auto-correlation analysis showed that the obtained VODCA-L-band VOD has a lower temporal autocorrelation than the original LPRM-SMAP VOD (Fig. 7). This indicates that the level of noise in L-band was increased with the merging. Hence for L-band, the merging results in a lower-quality data set. In addition, the low density of observations in LPRM-SMOS causes a highly unbalanced temporal coverage of VOD values (Fig. 8). Given this unbalanced data coverage there is a high risk that users might wrongly use this dataset for e.g. trend analysis. However, we are convinced that there is a large potential to produce a more reliable multi-sensor merged L-band data set in the future, e.g. by using alternative L-band retrievals from SMOS-IC or MTDCA. Yet, the current VODCA methodology cannot be easily applied to L-band data. Hence alternative blending approaches first need to be thoroughly assessed, which goes beyond the scope of the current paper.

We will add a paragraph in the introduction explaining in detail the reasoning behind not producing VODCA-L. We will also include figures 6 and 7 in the supplement for that purpose and expand the discussion regarding the possibility of an L-band product. We will also take care to talk specifically of LPRM-L-VOD and not L-VOD in general.

6. Page 4, line 10. A reference to the tau-omega model is needed. Please include: T. Mo, B. Choudhury, T. Schmugge, and T. Jackson, "A model for microwave emission from vegetation-covered fields," J. Hydrol., vol. 184, no. C13, pp. 101–129, Dec. 1982

We already referenced this paper just before at line 5 but we agree that it is not prominently enough. We changed the sentence:

"LPRM v6 (van der Schalie et al., 2017; Owe et al., 2008; Meesters et al., 2005) retrieves soil moisture and VOD at the same time from vertical and horizontal polarized microwave data and is based on a radiative transfer model first proposed by Mo et al. (1982).
The model assumes..."

7. Table 1: It would be interesting to add ascending and descending times for each sensor as well as their incidence angles, spatial and temporal resolutions. The authors could perhaps add a little discussion on the impacts of mixing the different times and observation geometries (spatial resolution, incidence angle, etc).

This is a very good idea, we are going to adjust the table and add a literature-heavy paragraph about this topic.

– Updated table with ascending/descending information.

– Expanding subsection "3.3 Merging" to discuss the effect of mixing sensors more in depth

8. Page 4, line 24: it is unclear how the different data sets can be accessed (web-page?). Please, specify which ones are available and which ones are not (perhaps on Table 2 also).

Agreed, this is necessary information in a data set paper.
We added following paragraph:

[4 : 24] While LPRM v6 is not publicly available, older versions are available trough GFSC: https://disc.gsfc.nasa.gov/datasets/LPRM_AMSR2_D_SOILM3_001/summary

9. Page 9, Line 25. I understand AMSR-E is used as a reference for having the highest overlap. But perhaps AMSR-2 could also be chosen for being a more advanced instrument with improved capabilities, or also a modeled VOD could potentially be used. Please, include a discussion for this choice (or provide a reference) and why it was chosen over the alternatives.

AMSR2 has a very similar design and skills as AMSR-E but has only little overlap with the other data sets. We do not want to use modeled VOD as a reference as we want to stay as close to the observations as possible. Using modeled VOD may introduce biases, e.g. related to uncertainties in the forcing data set.

We added a reference to CCI SM which uses AMSR-E as reference for the same reason.

405

10. Page 6, line 1. Please, indicate how to access the ancillary data used in the corresponding subsection (LAI and VCF).
We added the links to it (they already were in the acknowledgements and references).

11. Page 6, line 4. What do the authors expect from the comparison of VOD and LAI? A rationale of why they chose LAI
410 over other indices (e.g. NDVI, EVI) and whether they expect a higher correlation with any of the specific VOD products is needed

It serves two purposes. Mainly, the lack of ground truth makes the validation of VOD data difficult. At field studies with different crop types, it has been shown that VOD is closely related to LAI (Sawada et al., 2016). LAI has been also used to assess
415 other VOD products from active sensors (Vreugdenhil et al., 2017). For the assessment of VOD, we prefer LAI in comparison to NDVI because NDVI saturates earlier at high biomass levels than LAI. However, we do not expect the correlation to be very high as, as you mentioned earlier, LAI is a measure of leaf area while VOD is related to vegetation water content. As the higher frequencies (Ku-band) are more sensitive to the vegetation canopy than the low frequencies (L-band), we expect higher correlations between Ku- and X-band VOD with LAI than between L-band VOD and LAI.

420

We expanded the motivation a bit to explain better why LAI was chosen.

12. Page 7, line 8: How is the VOD climatology from AMSR-E derived? Please provide details.

425 This paragraph is just a summary, the details are in the subsequent subsection. By climatology we mean that if a sensor observes a specific combination of earth surface properties, LPRM will derive a specific VOD value. However the use of the term climatology is truly confusing in this context.

Alternatively we propose:

430

[7 : 8] Second, bias between the different sensors is corrected for by scaling them to VOD from AMSR-E C-, X-, and Ku-band, respectively.

13. Page 7, line 26. I agree with the authors that negative VOD retrievals are physically impossible. However, they are most
435 probably linked to uncertainties/simplifications on the physical model used in the inversion, and their direct truncation may lead to erroneous trends for specific areas. One alternative could be to let the user truncate the values after temporally averaging the data set according to the needs of their study. This is the procedure followed for instance in the SMOS-IC product (Fernández-Moranet al., remote sensing, 2017). I would ask the authors to consider this option or at least, mention it in the

discussion.

440

We agree that your proposed method is the correct way to release a data set based on a single sensor, and we also thought about doing it this way. But the trouble is that we average multiple sensors and redistribute only the averaged values. If we average the negative values with positive values from other sensors, we are averaging two observations with very little confidence in one of those. This leads to a lower-quality average than if we just discarded the negative value. We also considered to keep negative values if no other sensor has a valid value at that date. But that would mean that the time series will be spliced with values that are both negative AND are the result of only one sensor and therefore are of a lot lower quality than the other values in the time series. We might do something like that in a future version, but there are many open questions to it; it would also require a large rework of the code.

450    14. Page 8, line 2. How different are the retrievals from the two C-band channels? Again the authors include a flag but this flag is not useful if it is not related to a quality indicator or any further recommendation is given.

We agree and calculated the temporal correlation between the C1 and C2 band, which are strongly correlated (Fig. 9)

455    Addition to manuscript (inclusive figure which will be in supplement):

[8 : 3] As the two C-bands are highly correlated, the use of one or the other has a minor impact on the quality of the dataset (Fig. 9).

460    15. Page 8, line 22. I infer from the text that there is a need to a new cdf-matching technique due to the presence of outliers in the VOD data set. Could this cdf-matching also improve the VOD data in Liu et al 2009, 2011? Could this cdf-matching improve the soil moisture merging within ESA CCI? The authors could perhaps elaborate on this, to better motivate the approach.

Yes, this is a general purpose method that can be used in similar situations. The the code will be included in https://github.
465    com/TUW-GEO/pytesmo in the future. We are right now (literally) evaluating this method in the ESA CCI SM product.

We will add a sentence to the manuscript mentioning the general usability of the method:

[8 : 22] We propose here improvements to this method to derive more robust scaling parameters that are not specific to VOD
470    data but rather should be generally applicable in similar situations.

16. Page 10, line 11. Does this happen very often? Could this be one aspect to improve to increase coverage?

A fraction in the order of $1/10^6$ to $1/10^8$ of values are lost this way, so only very little temporal coverage increase is to be gained here.

Added:

[10 : 11] These values are deemed unreliable and are removed. However, this occurs very rarely, only a fraction of about $1/10^6$ to $1/10^8$ of values are lost in this way.

17. Page 10, line 18. Have the authors tried with the median statistic? It is less sensitive to ouliers.

No, we have not but this is a good idea. As it will only make a difference if three or more sensors have a value at a certain date, the expected change would be small. And since we already released the data as a requirement to submit a manuscript here, it likely would not make a difference big enough to warrant a new release of its own. As such we will evaluate this for the new version.
We add a sentence to the discussion that taking the median instead of the mean is an option to be explored in the future.

18. Figure 3. I do not think this Figure is necessary.

This Figure has been changed many times, we were never content with it. As also reviewer #1 finds it "actively confusing" and it is redundant with the text, we agree and will happily get rid of it.

19. Figure 4. What is the dominant vegetation in the chosen pixel? Perhaps the author could also include an example of time series in which TMI is also used, for completeness.

It is in Austria, mostly farmland, taking a look at google maps it is about 20% forest, also the Danube flows through it. We would rather reduce the number of Figures as we already have a lot of them. We like the current location because it shows a case where AMSR2 cannot directly be scaled to AMSR-E.

Updated first sentence of figure 4 caption:
Example X-band time series (15.125°E, 48.125°N in Austria, mostly farmland with about 20% forest cover) at (a) different processing steps and (b) violin plot showing the effect of CDF-matching on the distribution of VOD.

20. Page 12, line 3. The authors could also perhaps refer to the L-band VOD spatial patterns, which are consistent and correlate well with canopy height (e.g. Konings et al., L-band vegetation optical depth and effective scattering albedo estimation

from SMAP, Remote Sensing of Environment, 2017).

510  Indeed L-band VOD is a good proxy for canopy height or biomass. We will mention this in the Conclusions.

21. Figure 6. It is hard to see the seasonal patterns. The authors could perhaps consider showing only the period 2002-2017 (or even shorter)

515  One of the main purposes of this Figure is to illustrate the temporal and spatial extent of the different VODCA products. Therefore we show the full temporal coverage.

22. Page 15, line 1. It is unclear how the authors measure the spatio-temporal coverage. Is Fig. 8 showing the fraction of days each month as stated in the label? The final temporal resolution shown in the Figure and referenced in the text above is
520  unclear.

We agree, the sentences describing the calculation were a bit convoluted. The calculation for each month and latitude is: "number of observations of all pixels of a latitude" / ("number of land pixels of a latiude" x "number of days in month").

525  We reworked the label to figure 8:
Hovmoeller diagrams showing for each latitude and month the fraction of days per month with observations. The number of observations of a latitude and month are counted and then divided by the number of days per month and the number of land grid points at that latitude.

530  23. Page 15, line 14. What do the authors understand by a "CDF-matching failure"? could it be for one specific reason (e.g. see comment #15 above), or several? please,be more specific.

CDF matching fails if not enough AMSR-E data is available to retrieve robust scaling parameters. The rules are explained in detail in section 3.2.4, "Practical implementation and exceptions". We agree that not having a reference here to that subsection
535  makes it hard for a reader to retrieve that information.

We will summarize the reason shortly together with a reference to section "Practical implementation and exceptions" in page 15, line 14.

540  24. Page 19, line 29. This advice is helpful but it could really be useful and applicable if converted into a criteria that contributes to a quality flag. There is clearly a need for a quality flag.

This is already included as a quality flag (bit-flag: 11), but we did not mention it in the text. Currently it is only described in the methods, page 10, line 6-7 ("The published data sets contain a flag indicating the matching method, allowing the user to remove the AMSR2 observations matched directly to AMSR-E if desired."

To make it clearer, we will mention the existence of this flag again on page 19, line 29. Also, as already previously mentioned, we will add a section to the appendix listing all flags, their meaning and their effects on the data quality.

25. Fig. 12. I would suggest to include subfigures f and g into a separate Figure, for clarity.

We concur, separating them by time span makes it clearer. We separated them.

26. Page 21, line 17. Only Ku band spans three decades, this sentence is a bit misleading.

True, this is not precise. How about:

[21 : 17] We present to the scientific community VODCA, three long-term VOD data sets spanning up to three decades that can be used to study dynamics in the biosphere.

27. Page 21, line 21. From section 4.2., it is unclear that the resulting VOD data sets provide observations "on a daily basis".

Agreed, better to describe actual temporal coverage and not base sampling frequency. How about:

[21 : 21] ... with the added benefit of having observations unaffected by cloud cover, allowing generally for more than 40% of days having a valid VOD value.

28. Page 21, line 24. The authors could perhaps consider mentioning at some point in the manuscript that their work is particularly relevant in the context of the prospect launch of the multi-frequency candidate mission Copernicus Microwave Imaging Radiometer (CIMR, www.cimr.eu)

While we agree with the sentiment, this would only fit in if we would have already talked about this extensively in a previous section. Each other sentence is summarizing a (sub)section, so even if we would mention CIMR before it would still not fit in the conclusions.

**References**

[revised manuscript text omitted]

---

## Author Response (AR2)

**Author comments to: The Global Long-term Microwave Vegetation Optical Depth Climate Archive VODCA**

Leander Moesinger[1], Wouter Dorigo[1], Richard de Jeu[2], Robin van der Schalie[2], Tracy Scanlon[1], Irene Teubner[1], and Matthias Forkel[1]

[1]Vienna University of Technology, Department of Geodesy and Geoinformation, Gußhausstraße 27-29, 1040 Vienna, Austria
[2]VanderSat, Wilhelminastraat 43A, 2011 VK Haarlem, The Netherlands

**Correspondence:** Leander Moesinger (Leander.Moesinger@geo.tuwien.ac.at, vodca@geo.tuwien.ac.at)

Formatting as follows:

Reviewers' comments

Reply to comments

[page:line]

5 [page:line] Added/changed parts to the manuscript

Titles

2.1 VOD data sets and 2.2.1 Liu et al VOD

For better readability please write out Vegetation Optical Depth in the titles

10 E.g. 2.2.1 Vegetation Optical Depth products from Liu et al, LiuVod

May be you change the order in the short name to VOD_Liu or similar, it is more familiar to readers with the variable name used in the first place

Yes, this does increase the readability

Vegetation Optical Depth is now written out in all titles and changed LiuVod to VOD_Liu on various occasions

15 Under 2.2.1 ….We expect that the data that are publicly available were subject to some temporal smoothing since the data are almost gap-free. Unfortunately …Better to phrase the sentence Starting with 'Unfortunately' differently, e.g. 'no details on it available in Liu et al 2011, 2015 and supplementary information.…

[6:8] Unfortunately, we could find no mention of this in either of the two papers and their supplementary information

[6:8] The smoothing is not described in Liu et al. (2011b, 2015) and supplementary information

3.2 Looks like 3.2.1. and 3.2.2 can be merged together to one chapter?

Agreed, those are better merged

Merged sections with only minor edits to bulk text

25     3.2.4 Practical implementation sound like a good title, dismiss 'exceptions' in 3.2.4

Agreed, changed it

Figure 15 figure caption: name the product name VCF before tree canopy, short vegetation, . . . .

Agreed, named VCF in caption

Figure 4, 9, 10, 11,12, 13,14,15 look good but hard for the reader to see details- Please enlarge all global distribution maps.

We increased the size of all these figures to the maximum width and increased legend font sizes for better readability

I like to use the ESA software SNAP for visualisation that usually reads well all different types of nc files. SNAP - old and

35   newest software version could not read the VODCA file from the product series Band C, I did not test the other frequency

products,I could read the C-Band data in r. could you investigate this issue

Thank you for the feedback, we were not aware of that. We were able to reproduce the issue, but did not have time yet to

look into the problem in detail. We also are not able to simply exchange the data already uploaded to Zenodo as it is connected

to a DOI. But we will take care of issue when we update VODCA with the 2019 data in January 2020. For the time being,

40   Panoply is able to open and plot the images for a quick look.

**Main changes to: The Global Long-term Microwave Vegetation Optical Depth Climate Archive VODCA**

Leander Moesinger[1], Wouter Dorigo[1], Richard de Jeu[2], Robin van der Schalie[2], Tracy Scanlon[1], Irene Teubner[1], and Matthias Forkel[1]

[1]Vienna University of Technology, Department of Geodesy and Geoinformation, Gußhausstraße 27-29, 1040 Vienna, Austria
[2]VanderSat, Wilhelminastraat 43A, 2011 VK Haarlem, The Netherlands

**Correspondence:** Leander Moesinger (Leander.Moesinger@geo.tuwien.ac.at, vodca@geo.tuwien.ac.at)

The suggestions have been implemented, but other than that no changes have been made save some spelling errors and a formatting problem with Table 1.

[revised manuscript text omitted]

---

## Author Response (AR3)

**Author comments to the second revision of: The Global Long-term Microwave Vegetation Optical Depth Climate Archive VODCA**

Leander Moesinger[1], Wouter Dorigo[1], Richard de Jeu[2], Robin van der Schalie[2], Tracy Scanlon[1], Irene Teubner[1], and Matthias Forkel[1]

[1]Vienna University of Technology, Department of Geodesy and Geoinformation, Gußhausstraße 27-29, 1040 Vienna, Austria
[2]VanderSat, Wilhelminastraat 43A, 2011 VK Haarlem, The Netherlands

**Correspondence:** Leander Moesinger (Leander.Moesinger@geo.tuwien.ac.at, vodca@geo.tuwien.ac.at)

Formatting as follows:

Reviewers' comments

Reply to comments

[page:line]

5 [page:line] Added/changed parts to the manuscript

some minor formal requirements:

Thank you for editing the figures of the global distribution maps

– the font size of the subtitles a), b) c) is inconsistent – e.g. too large for figure 9

10 In the Supplement please also enlarge the global distribution maps figures 2 to 6 and provide a consistent font size for the subtitles in the manuscript and the supplement

We agree, the size of fonts and figure sizes was inconsistent

15 – Reduced subtitle font size of figure 9, nothing else was changed in the main document.

– Increased size of map-style supplementary figures and in the process split figure 7 into figures 7-9 because it did not fit anymore on one page.

– Adjusted style and font sizes of colorbar text for supplementary figures 1-3

– Adjusted font size of subfigure titles of supplementary figure 3

20 Data publication in Zenodo:

Please add the information on the issue that the products are currently not readable in SNAP to the abstract and the information that a new update VODCA with the 2019 data will be available in January 2020.

This will become a new version of the data set including 2019 that should be published under the same DOI, but with another

version number, Zenodo is supporting versioning.

25

 Added to Zenodo description:

Currently there is an issue with opening the file using ESA SNAP. As an alternative Panoply can be used to quickly visualize the data.

An update of VODCA, addressing this issue and potentially including an extension of the dataset, is foreseen to be published

30 on Zenodo early 2020.

**Main changes to: The Global Long-term Microwave Vegetation Optical Depth Climate Archive VODCA**

Leander Moesinger[1], Wouter Dorigo[1], Richard de Jeu[2], Robin van der Schalie[2], Tracy Scanlon[1], Irene Teubner[1], and Matthias Forkel[1]

[1]Vienna University of Technology, Department of Geodesy and Geoinformation, Gußhausstraße 27-29, 1040 Vienna, Austria
[2]VanderSat, Wilhelminastraat 43A, 2011 VK Haarlem, The Netherlands

**Correspondence:** Leander Moesinger (Leander.Moesinger@geo.tuwien.ac.at, vodca@geo.tuwien.ac.at)

Adjusted subtitle font size of figure 9, made numerous smaller changes to supplementary figures and updated description on Zenodo. We are not providing marked-up manuscript changes for this revision as no text changes have been made.